# Multiple objective immune wolf colony algorithm for solving time-cost-quality trade-off problem

**Guanyi Liu**[1], **Xuemei Li**[1]*, **Khalid Mehmood Alam**[2]

**1** Department of Economics and Management, Beijing Jiaotong University, Beijing, Beijing, China, **2** China Study Centre, Karakoram International University, Gilgit -Baltistan, Pakistan

* 467713868@qq.com

**Data Availability Statement:** All relevant data are within the manuscript and its Supporting Information files.

**Funding:** The author(s) received no specific funding for this work.

## Abstract

The importance of the time-cost-quality trade-off problem in construction projects has been widely recognized. Its goal is to minimize time and cost and maximize quality. In this paper, the bonus-penalty mechanism is introduced to improve the traditional time-cost model, and considering the nonlinear relationship between quality and time, a nonlinear time-cost quality model is established. Meanwhile, in order to better solve the time-cost-quality trade-off problem, a multi-objective immune wolf colony optimization algorithm has been proposed. The hybrid method combines the fast convergence of the wolf colony algorithm and the excellent diversity of the immune algorithm to improve the accuracy of the wolf colony search process. Finally, a railway construction project is taken as an example to prove the effectiveness of the method.

## Introduction

With the continuous advancement of global urbanization and the increasing demand for infrastructure construction in cities, construction planning is a basic and challenging task in construction project management. How to plan the project's progress scientifically and reasonably has been a hot issue for a long time. At the project planning stage, it is necessary to meet three main objectives, namely, deliver the project according to the time, cost and quality requirements specified in the contract. However, in the project construction stage, customers need to speed up the project or ensure delivery within the planned date, which requires the project manager to adjust the relationship between time, cost, and quality. However, this relationship is contradictory and related [1]. In general, in order to shorten the construction period, project managers will design new alternatives for each construction activity and re-formulate the schedule, but this will affect the time, cost, and quality of the construction project. For example, by increasing large construction equipment and workers' labor time, the duration of construction activities can be significantly reduced, but the total cost of construction will also increase, and there may be project quality problems. Therefore, it is of great significance to study the trade-off between time, cost, and quality.

**Competing interests:** The authors have declared that no competing interests exist.

The optimization goal of traditional construction project planning is to analyze the trade-off between time and cost in the project to minimize the cost. The classical critical path method is regarded as an acceptable technology to determine the effective cost in construction project planning [2]. This method analyzes the impact on the cost according to the duration of activities on the critical path and the priority relationship between activities. After the introduction of the critical path method, many researchers began to discuss the problem of time-cost trade-off. For example, Fulkerson [3] proposed a linear programming problem to calculate the project time cost trade-off problem composed of multiple individual tasks. These tasks have a related collapse time and normal time, and the cost of project activities varies linearly between these extreme times. Robinson [4] proposed a dynamic programming method to solve the time-cost trade-off problem in project management with available models, which realizes the optimal allocation between activities with an arbitrary cost time function. Reda R and Carr RI [5] compared the practical method of time-cost trade-off (TCT) with the computerised TCT method according to the actual construction planners and considered that the correlation of construction activities is very important, and the normal computerized TCT technology is conceptually wrong for construction. De P et al. [6] considered the importance of discrete time-cost trade-off problems, proposed a solution of network decomposition/reduction, and emphasized the necessity of developing and evaluating effective procedures to solve general time-cost trade-off problems. Laslo Z [7] described the stochastic expansion model of time-cost trade-off problem of the critical path method and constructed four basic formulas of time-cost trade-off model based on different assumptions, avoiding the infeasibility of additional expert group evaluation for each possible performance speed. Chen SP and Tsai MJ [8] proposed a new method for time-cost trade-off analysis of projects network in a fuzzy environment. The fuzziness of the minimum total collision cost provides more information for time-cost trade-off analysis in project management. Al Haj and El-Sayegh [9] proposed a nonlinear integer programming model to solve the time cost optimization problems considering the influence of total floating charge loss. Alavipour and Arditi [10] integrated the optimization of project financing costs into the analysis of time-cost trade-off and proposed an integrated model. Sonmez, Aminbakhsh, Atan [11] proposed a new non-critical job rescheduling method to narrow the gap between time-cost optimization research and practice.

However, with the in-depth study of project construction, it is found that decision-makers still need to consider project quality when adjusting the relationship between construction period and cost. When dealing with large-scale projects, it is difficult for decision makers to find the best solution with the shortest time, the lowest cost, and the highest quality with their own experience. Therefore, quality has become an indispensable and important factor. Taking quality maximization as the third optimization goal leads to the problem of time-cost quality tradeoff. As a result, a large number of researchers have devoted themselves in recent years to solving the problem of time-cost-quality tradeoff. Babu and Suresh [12] first developed a linear programming model of time, cost, and quality, and proposed that the collapse time could affect project quality. Khang and Myint [13] took cost and quality as affine functions of duration and took the average value of activity quality as the total quality of the project. In addition, they claimed that the linear assumption between quality and time was questionable. El-Rayes and Kandil [14] proposed a multi-objective optimization model to transform the traditional two-dimensional time-cost trade-off analysis into a three-dimensional time-cost quality trade-off analysis. Tareghian and Taheri [15] added the quality factor to the discrete-time cost model and developed a programme to solve the time-cost quality problem. Zhang H and Xing F [16] considered that the quality could not be collected and recorded by accurate numbers, described the time, cost, and quality by fuzzy numbers, and evaluated the selected construction method by using a fuzzy multi-attribute efficiency method combined with constrained fuzzy

arithmetic operation. Kim J, Kang C, Hwang I [17] considered the potential quality loss cost associated with rework or modification that may be caused by excessive collapse activities and proposed a mixed integer linear programming model. Nabipoor Afruzi, et al. [18] constructed a discrete-time cost-quality trade-off model with limited pattern identification resources considering the resource constraints in the current project. The model requires that each activity has multiple modes to execute, and in each execution mode, each activity can be completed in normal or crash mode. Zhang L, Du J, Zhang S [19] introduced reward and punishment and established a new time-cost quality integrated optimization model. In addition, a new quality model called the Quality Performance Index (QPI) was established to describe the reliability of the system. Tran DH, Cheng MY, Cao MT [20] adopted the definition of personal quality, considered multiple execution modes in which each mode in each activity has an estimated duration, and calculated the quality and cost according to their respective duration functions. Jeunet J, Bou Orm M [21] consider personal quality constraints, priority relationships, non-preemption and resource availability. A mixed integer linear programming model is developed to optimize temporary work and overtime, to speed up the project progress while considering quality and productivity.

In addition to constructing the target mathematical model, the time-cost quality trade-off problem also needs excellent optimization methods to solve. Because TCQTP belongs to a multi-objective optimization problem, many researchers devote themselves to developing different optimization algorithms to solve the problem. At present, common optimization algorithms include particle swarm optimization algorithm, fast non-dominated sorting genetic algorithm with elite retention strategy (NSGA-II), ant colony algorithm, simulated annealing algorithm, and so forth [22–24]. For example, Fallah-Mehdipour [25] compared the multi-objective particle swarm optimization algorithm (MOPSO) with the non-dominated sorting genetic algorithm (NSGA-II) and proved that NSGA-II can determine the optimal scheme more successfully than the MOPSO algorithm on the issues of time cost trade-off (TCTO) and time cost quality tradeoff (TCQTO). Fu F, Zhang T [26] proposed a hybrid leapfrog algorithm, that combines the crossover operator of a genetic algorithm with local search based on substitution in the process of leapfrog. The performance of the algorithm is verified by taking the construction project of a railway overpass as an example.

Nadimi-Shahraki, et al. [27] proposed an improved moth-flame optimization (I-MFO) algorithm, introduced the adapted wandering around search (AWAS) strategy to escape the local optimal solution, evaluated the performance of the proposed algorithm through a benchmark function, and compared it with other well-known metaheuristic algorithms. Finally, I-MFO is used to solve practical mechanical engineering problems. In Khodadadi N, Azizi M, Talatahari S, Sareh P [28] considering the multi-objective optimization problem of multiple performance indicators inspired by the principle of crystal structure formation, a meta heuristic algorithm called the Crystal Structure algorithm (Crystal) is proposed, and the algorithm is evaluated. The results show that the algorithm provides excellent results when dealing with multi-objective problems. To Nadimi-Shahraki MH, et al. [29] reduce the high selection pressure and low diversification of GWO algorithm, a grey wolf optimizer based on gaze cue learning (GGWO) is proposed. The algorithm introduced two strategies: neighbor gaze cues learning (NGCL) and random gaze cues learning (RGCL) to enhance the optimization ability. The GGWO algorithm is compared with other algorithms. The results show that the GGWO algorithm has better performance. Based on Azizi M, et al. [30], Multi-Objective Atomic Orbital Search (MOAOS) is proposed to solve multi-objective optimization problems. By using MOAOS to evaluate benchmark problems ZDT and DTLZ, the results show that this algorithm can produce superior or similar results compared with other meta-heuristic methods. Khodadadi N, Abualigah L, Mirjalili S [31] made appropriate changes to the stochastic

paint optimizer (SPO) and proposed a multi-objective stochastic paint optimizer (MOSPO) to solve multi-objective optimization problems. Compared with MOPSO MSSA and MOALO, MOSPO had high convergence and excellent Pareto front results in dealing with multi-objective engineering problems.

It is not difficult to conclude from the above literature review that TCQT has been widely addressed and effectively resolved. However, the existing TCQT mathematical model has not considered the impact of reward and punishment factors on construction projects, and intelligent optimization algorithms with better solution performance need to be proposed. Thereby, the contributions of this study are stated as follows: (1) clarify the nonlinear relationship between time and bonus-penalty cost, a new bonus-penalty cost model is presented. Therefore, a new multi-objective mathematical model of the time-cost quality trade-off problem with several equality and inequality constraints is established. (2) A new multi-objective immune wolf swarm algorithm has been developed to solve the trade-off problem of time, cost, and quality. The proposed MOIWCA can solve the problem that the traditional wolf colony algorithm is easy to fall into a local optimal solution. At the same time, in order to improve the search speed and optimization performance of the algorithm, the cross operation and immune operation in the immune algorithm are improved. The algorithm can provide a representative and managed Pareto set for the time-cost-quality trade off problem. (3) To make sure that the proposed MOIWCA works and can be used, we use a high-speed railway construction project as a case study to show that the proposed method is better.

The rest of this paper is organized as follows. In section 2, the nonlinear relationships of time, cost, and quality are analyzed, and the mathematical models of time, cost, and quality are given, respectively. Section 3, the multi-objective immune wolf colony algorithm is proposed. In section 4, an experimental test and a case study of a railway construction project are shown to verify its effectiveness. Finally, the main conclusions and future work of the paper are presented.

## Model mathematical optimization

### Mathematical optimization model of the minimum time

**Method of the critical path.** The critical path method (CPM) is defined as the technology to analyze and obtain the critical path of a project without considering any resource constraints and given the construction time, logical relationship and other time constraints [32]. By using forwards and backwards methods, CPM identifies the possible relationships among the activities in the project, namely the start-to-start (SS), the finish-to-finish (FF), the start-to-finish (SF), and the finish-to-start (FS) relations. In the PDM diagram, the path with the longest operation duration in the schedule network is the critical path, which determines the shortest possible project duration.

The start-to-finish (SF) is not considered in this paper because it means that the beginning of an activity requires the realization of another activity. If the latter is delayed, there is no feedback to the former at all. When using the CPM to organise the activities sequence, make sure that each activity has at least one predecessor activity and one successor activity (except for the start and end of the project's activities). Meanwhile, to keep the schedule dynamics and effectiveness, each activity has FS or SS in its predecessor logic relationship and has FS and FF in its successor.

In addition, lead and lag of time are involved in the generalized precedence relationships (GPR) between activities [33]. GPR can specify a minimum or maximum time lag between any pair of activities. The minimum time lag means that an activity can only be started (finished) if the predecessor activity has already been started (finished) within a period of time. The

maximum time lag means that an activity should have recently started (finished) within a specific period after the start (finish) of other activities [34].

In this paper, the minimum time of a project is the sum of the times of all activities on the critical path. The constraints are the precedence relationships between the activities.

**Project minimum time and constrains.**   As mentioned before, we use the CPM to determine the precedence relationships among the activities in the project to determine the constraints in the mathematical model of the minimum project time [35].

Under the given technology and resource conditions, the minimum and maximum activity realization times can be calculated for each activity on the project. In the activity $i$, the minimum activity realization time is represented as $t_i^{min}$, and the maximum activity realization time is represented as $t_i^{max}$. In addition, general time $t_i^e$ is introduced. Regardless of policies, weather, and other special circumstances, the time relationship of activity $i$ can be expressed by the following inequation:

$$t_i^{min} \le t_i \le t_i^{max} \qquad i = 1, 2, 3, \ldots, n \tag{1}$$

where $n$ is the number of activities on the project.

Then we respectively define the minimum project time $T_{min}$ and the maximum project time $T_{max}$, and the total project realization time $T$. The value of $T$ is equal to the sum of time of all activities on the critical path. Therefore, the relationship is described by the following inequation:

$$T_{min} \le T \le T_{max} \tag{2}$$

The time function is defined as the sum of the duration taken by all the activities in the critical path of the project while maintaining relationships between the predecessor and successor activities. The minimization of project total time is shown as follows:

$$Min \; T = \sum_{i=1}^{n} \sum_{j=1}^{m_i} \varphi_i t_{ij} \tag{3}$$

where $T$ is the project total time; $\varphi_i$ is a binary variable that is equal to 1 if the activity $i$ is selected on the critical path, otherwise is equal to 0; $t_{ij}$ is the duration of option $j$ for activity $i$; $m_i$ is the number of subcontracting option for activity $i$.

As mentioned before, between the activities in the PMD diagram, there are the start-to-start (SS), the finish-to-finish (FF), and the finish-to-start (FS) relationships. These precedence relationships are used as constraints for the minimum time. Assume that activity $i$ is the predecessor activity of activity $j$, activity $k$ is the successor activity of activity $j$, and lead and lag of time are involved in the precedence relationships. Therefore, we have three different relationships as follows:

For the relationship of "finish-start" is shown in Fig 1.

$$s_j^E - s_i^E \ge (dur_i - \omega t_i) + FS_{ij}^{min} \tag{4}$$

$$f_k^L - f_j^L \ge (dur_k - \omega t_k) + FS_{jk}^{min} \tag{5}$$

$$f_j^L - f_j^E \ge dur_j - \omega t_j \tag{6}$$

where $s_i^E$ is the earliest start time of activity $i$; $s_j^E$ is the earliest start time of activity $j$; $f_j^L$ is the latest time of activity $j$; $f_k^L$ is the latest time of activity $k$; $dur_i$ is the duration of activity $i$; $dur_j$ is the duration of activity $j$; $dur_k$ is the duration of activity $k$; $FS_{ij}^{min}$ is the minimal lag/lead between two activities of $i$ and $j$ for FS relationships; $FS_{jk}^{min}$ is the minimal lag/lead between two activities

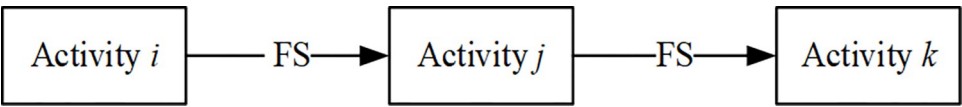

**Fig 1. The relationship of "finish-start".**

of $j$ and $k$ for FS relationships; $t$ is the reduction time of the activity; $\omega$ is a binary variable that is equal to 1 if the activity is reduced $t$, otherwise is equal to 0.

For the relationship of "start-start" is shown in Fig 2.

$$s_j^E - s_i^E \geq SS_{ij}^{min} \tag{7}$$

$$f_k^L - s_j^E \geq (dur_k - \omega t_k) + SS_{jk}^{min} \tag{8}$$

where $SS_{ij}^{min}$ is the minimal lag/lead between two activities of $i$ and $j$ for SS relationships; $SS_{jk}^{min}$ is the minimal lag/lead between two activities of $j$ and $k$ for SS relationships.

For the relationship of "finish-finish" is shown in Fig 3.

$$f_j^L - s_i^E \geq (dur_i - \omega t_i) + FF_{ij}^{min} \tag{9}$$

$$f_k^L - f_j^L \geq FF_{jk}^{min} \tag{10}$$

where $FF_{ij}^{min}$ is the minimal lag/lead between two activities of $i$ and $j$ for FF relationships; $FF_{jk}^{min}$ is the minimal lag/lead between two activities of $j$ and $k$ for FF relationships.

## Mathematical optimization model of the minimum cost

**Nonlinear bonus-penalty theory and the cost model of bonus-penalty.** In order to complete the project ahead of schedule, the decision-maker encourages the contractor to reduce the project total time through the use of economic incentives; and in order to guarantee the completion of the project on the contract date, the decision-maker discourage the contractor from delaying the project time by means of economic penalties. Through the analysis of relevant literature [36–38], it can be seen that the existing research on bonus-penalty models of project duration generally adopts linear mathematical models, which ignore the problem of insufficient bonus-penalty and the risk of quality and safety. Based on the nonlinear reward model proposed [39], this paper puts forward the nonlinear bonus-penalty theory, which implies to that decision-makers have psychological expectations for the reduction or delay of project duration under the external operating environment. When the decision-maker decides to reduce project duration by economic incentive, the incentive intention shows a trend of rising first and then falling; when the decision-makers find that the project duration is extended,

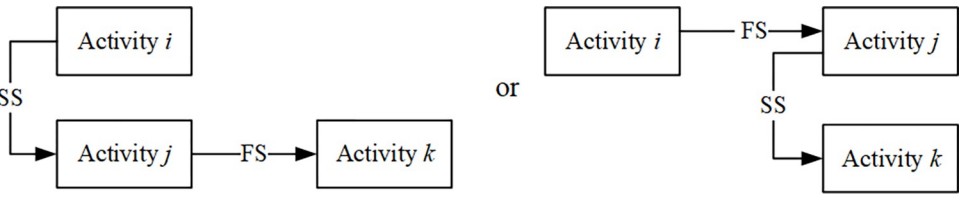

**Fig 2. The relationship of "start-start".**

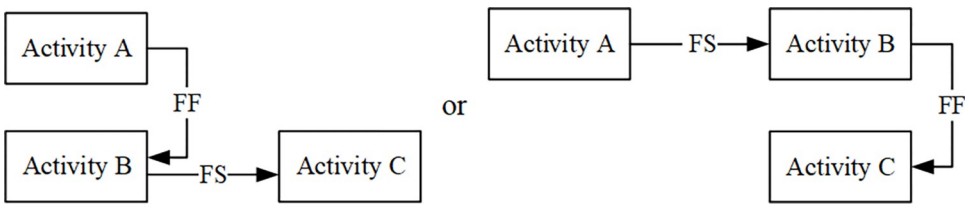

**Fig 3. The relationship of "finish-finish".**

the penalty intention shows a trend of slow first and then quick. The specific bonus-penalty intention curves are shown in Figs 4 and 5.

According to the above analysis, there is an N-type nonlinear relationship between the time reduction of a project and the bonus, which is manifested as the amount of bonus increasing marginally first and then decreasing marginally with the increase in time reduction of the project. By simulating the N-type curve form, the nonlinear characteristics of bonus for the time reduction of a project can be approximated by the following model:

$$C_{bonus} = C_{bonus}^{max} \left[ arctan(t_{actual} - t_{contract} - t_e) + \frac{\pi}{2} \right] \tag{11}$$

where $C_{bonus}$ is the cost of bonus; $C_{bonus}^{max}$ is the maximum cost of bonus; $t_{actual}$ is the actual realization time of project; $t_{contract}$ is the required time of the project contract; $t_e$ is the expected time reduction of the project.

There is an L-type nonlinear relationship between project time delay and penalty, which manifests as the amount of penalty increasing steadily and then marginally as project time delay increases. By simulating the L-type curve form, the nonlinear characteristics of the penalty for the time delay of a project can be approximated by the following mathematical model:

$$C_{penalty} = \begin{cases} \sum_{k=1}^{t_n} \varphi t_k \\ C_{penalty}^{t_n} + e^{t_{actual} - t_{contract} - t_n} - 1 \end{cases} \tag{12}$$

where $C_{penalty}$ is the cost of penalty; $t_k$ is the number of time delay of project ($k = 1, 2,\ldots, t_n$); $t_n$ is the maximum number of expected time delay of project; $C_{penalty}^{t_n}$ is the cost of penalty at the $t_n$.

**Project minimum cost and constrains.** In addition to reward and punishment costs, total project realization costs include direct and indirect project costs. The direct cost is directly related to the time of each activity of the project, mainly including the cost of labor, materials, construction machinery, and other resources. The indirect costs are related to the total duration of the project and include mainly taxes, project operating costs, personnel management fees, and other indirect costs.

The direct cost is closely related to the duration of each activity of the project, and the duration of the activity will affect the direct cost of the activity. Suppose that activity $i$ is one of the activities in the project, the minimum activity realization time is represented as $t_i^{min}$, the maximum activity realization time is represented as $t_i^{max}$, and the general activity realization time is represented as $t_i$. Accordingly, the minimum cost of activity $i$ is represented as $C_{i,d}^{min}$, the maximum activity cost is represented as $C_{i,d}^{max}$, and the general activity cost is represented as $C_{i,d}$.

The time of activity is reduced, we need to put in extra resources for the activity to succeed. This contributes to the direct cost of activity increasing, and according to practical engineering experience, with the continuously reducing the duration of the activity, the direct cost is rising, but the speed of rise is slowing gradually. Therefore, the link between the duration of activity

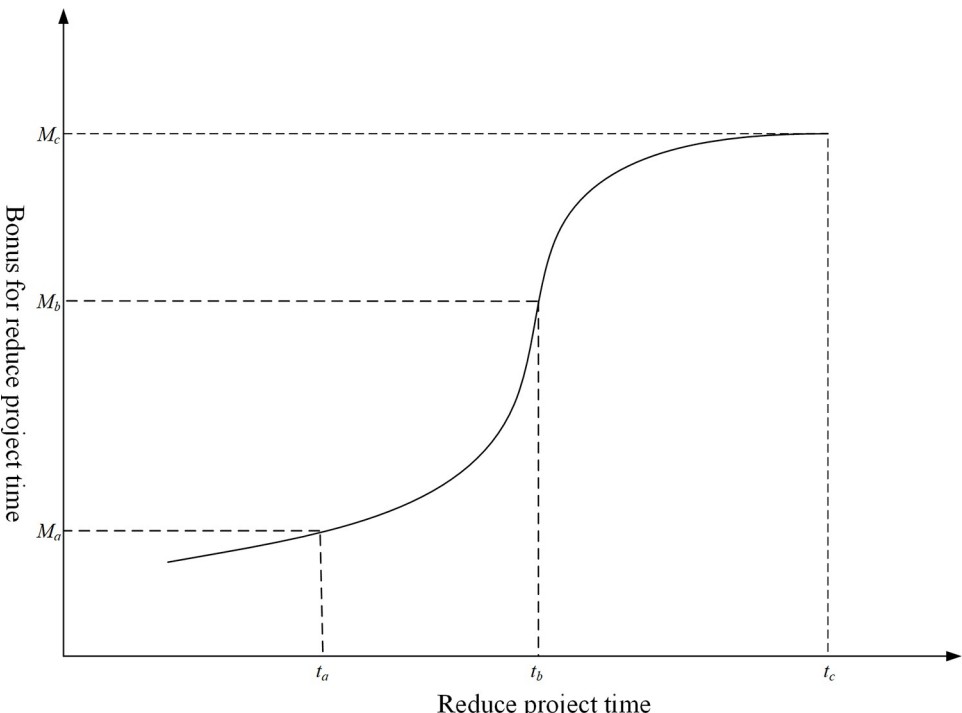

**Fig 4. Bonus of duration reduction graph.**

and direct cost is nonlinear. The relationship of nonlinear is shown in Fig 6. In this paper, we use the Lagrange quadratic interpolation formula to establish a mathematical model of the direct cost:

$$CD = \sum_{i=1}^{n}[CD_i^{min} \cdot k_i^{min}(t_i) + CD_i^c \cdot k_i^c(t_i) + CD_i^{max} \cdot k_i^{max}(t_i)] \tag{13}$$

$$k_i^{min} = \frac{(t_i - t_i^c)(t_i - t_i^{min})}{(t_i^{min} - t_i^c)(t_i^{min} - t_i^{max})} \tag{14}$$

$$k_i^c = \frac{(t_i - t_i^{min})(t_i - t_i^{max})}{(t_i^c - t_i^{min})(t_i^c - t_i^{max})} \tag{15}$$

$$k_i^{max} = \frac{(t_i - t_i^{min})(t_i - t_i^c)}{(t_i^{max} - t_i^{min})(t_i^{max} - t_i^c)} \tag{16}$$

Unlike the relationship between the direct cost and duration, it is difficult to show the impact in the change of duration of activity in the project on the indirect cost. We consider the relationship between the indirect cost and total time. According to the practical engineering experience, with the continuously increasing duration of the activity, the growth speed of indirect costs is slowing. This is because some indirect costs can be put into multiple uses at a time, such as the cost of management model formulation. There is no need to repeatedly put in. Therefore, the link between the total time of the project and direct cost is nonlinear, the relationship of nonlinear is shown in Fig 7. In this paper, we use the Lagrange quadratic

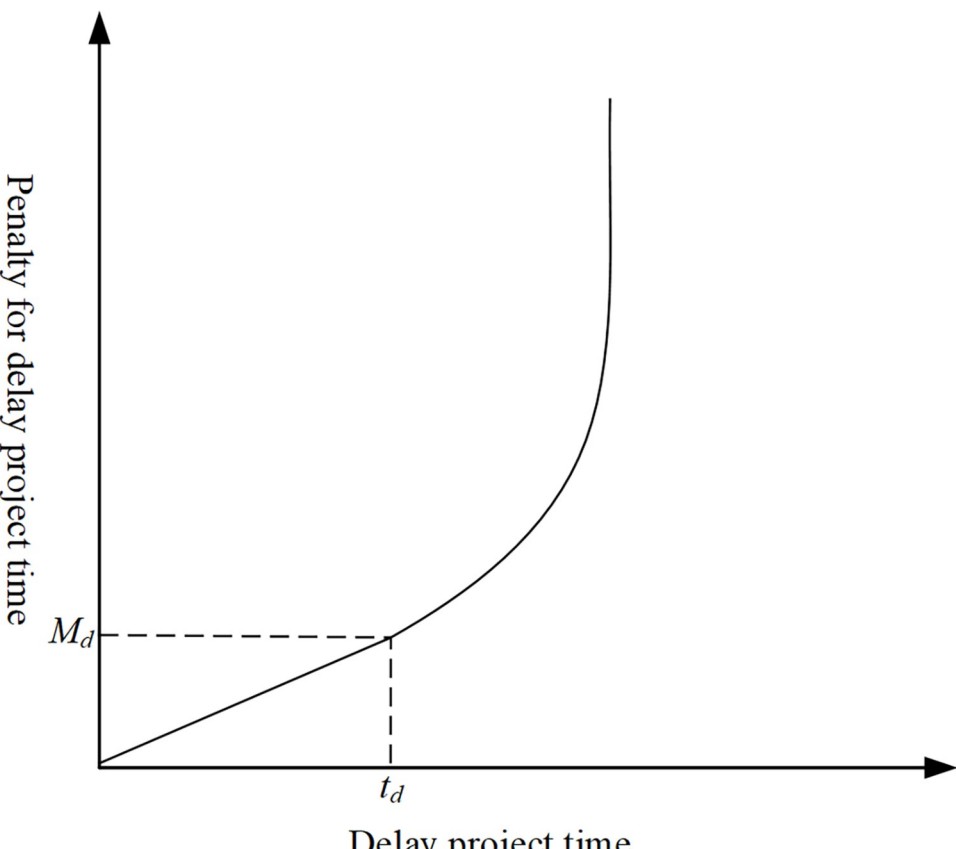

**Fig 5. Penalty of duration delay graph.**

interpolation formula to establish a mathematical model of the indirect cost:

$$CID = CID^{min} \cdot k^{min}(T) + CID^c \cdot k^c(T) + CID^{max} \cdot k^{max}(T) \tag{17}$$

$$k^{min} = \frac{(T - T^c)(T - T^{min})}{(T^{min} - T^c)(T^{min} - T^{max})} \tag{18}$$

$$k^c = \frac{(T - T^{min})(T - T^{max})}{(T^c - T^{min})(T^c - T^{max})} \tag{19}$$

$$k^{max} = \frac{(T - T^{min})(T - T^c)}{(T^{max} - T^{min})(T^{max} - T^c)} \tag{20}$$

The minimization of total cost is shown as follow:

$$Min\ C = CD + CID + C_{bonus} + C_{penalty} \tag{21}$$

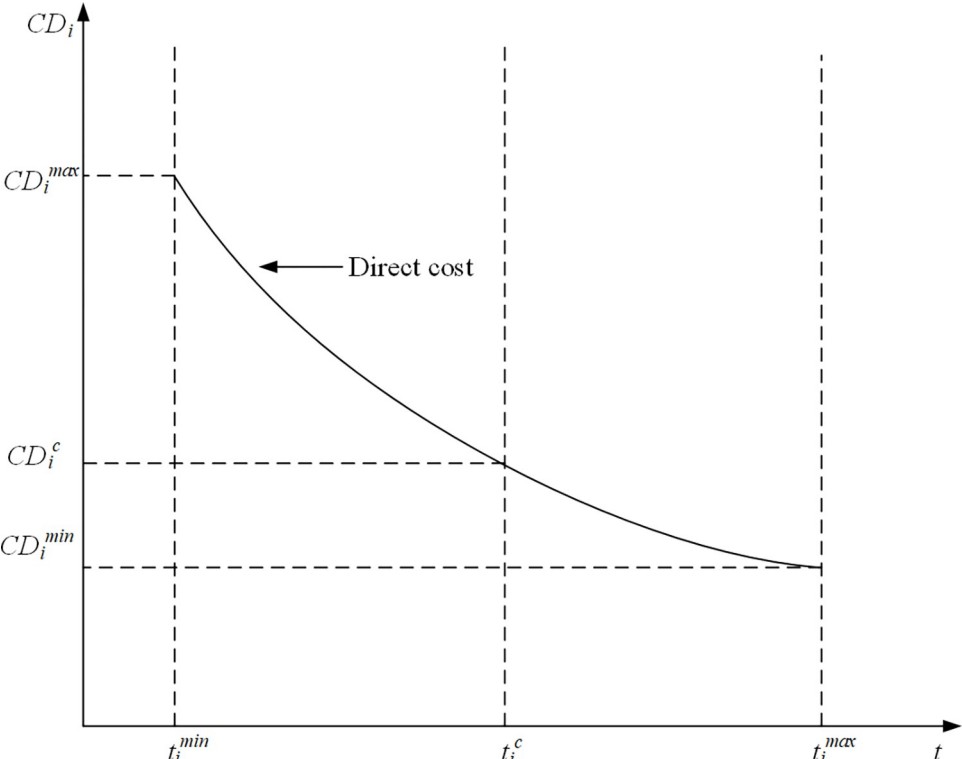

**Fig 6. The curve of the relationship of time- direct cost.**

## Mathematical optimization model of the maximum quality

In the process of project construction, quality has become an important factor for project managers to trade-off and make decisions. Since a project includes various resources, such as materials, machines, and labor, the overall quality of the project depends on the quality of each activity. The relationship between the quality and duration of activities is complex. In order to find out how time affects quality, we propose the following assumptions:

1.  The overall quality of the whole project depends on the quality of each activity.

2.  Appropriate extension of activity time will improve the quality level.

3.  At a certain point in time, the activity quality reaches the highest level; after this point, the extension of time will lead to the decline of quality level. Because in the activities of some construction projects, the relationship between activity time and quality is not linear.

Based on the above assumptions, we define the overall project quality as the integration of the quality of all activities. We define the quality performance index (QPI) as the level of quality that the contract has reached in a certain amount of time [20]. We define $QPI_i$ as the quality of a single activity as follows:

$$QPI_i = a_i t_i^2 + b_i t_i + c_i \qquad (22)$$

where $QPI_i \in [0, 1]$, $i = 1, 2, \ldots, n$; $t_i$ is the during of activity $i$, with $t_i > 0$; $a_i$, $b_i$, $c_i$ are the coefficients decided by the quadratic function (Fig 8). Fig 8 shows that $SD_i$, $BD_i$, $LD_i$ are the shortest duration, best duration, longest duration of activity $i$, and $BD_i$ corresponds to the maximum value of $QPI$. The parameters can be provided by the deterministic data from the case study.

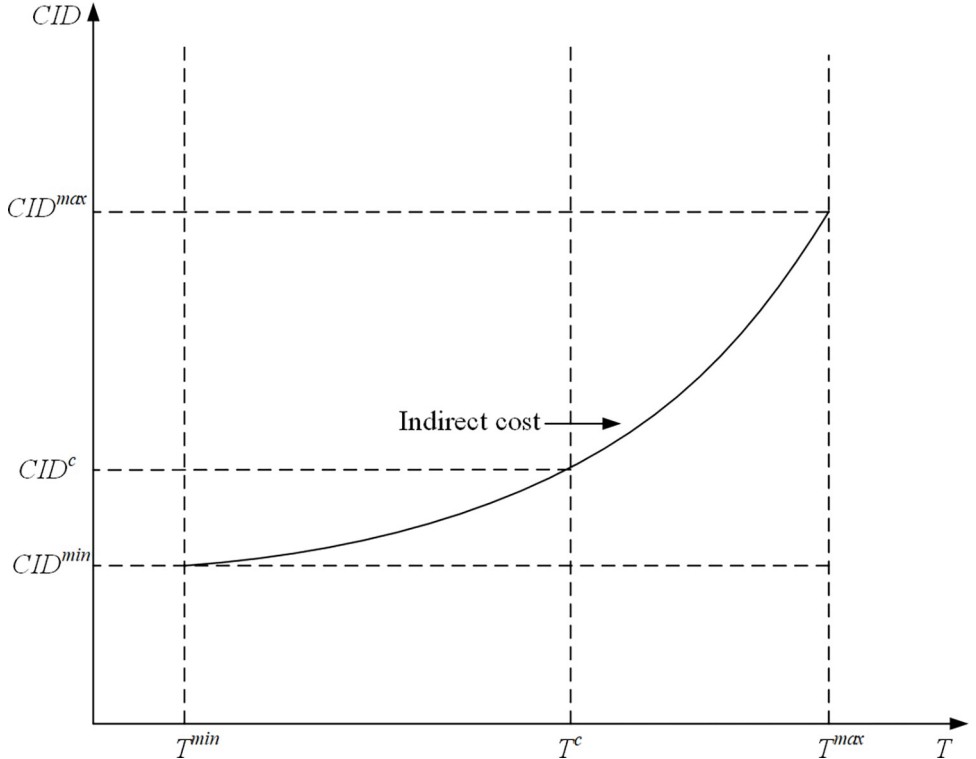

**Fig 7. The curve of the relationship of time- indirect cost.**

Based on our prior investigation of real construction projects and the results of statistical analysis, $BD_i$ can be determined by the following equation:

$$BD_i = SD_i + 0.6774(LD_i - SD_i) \tag{23}$$

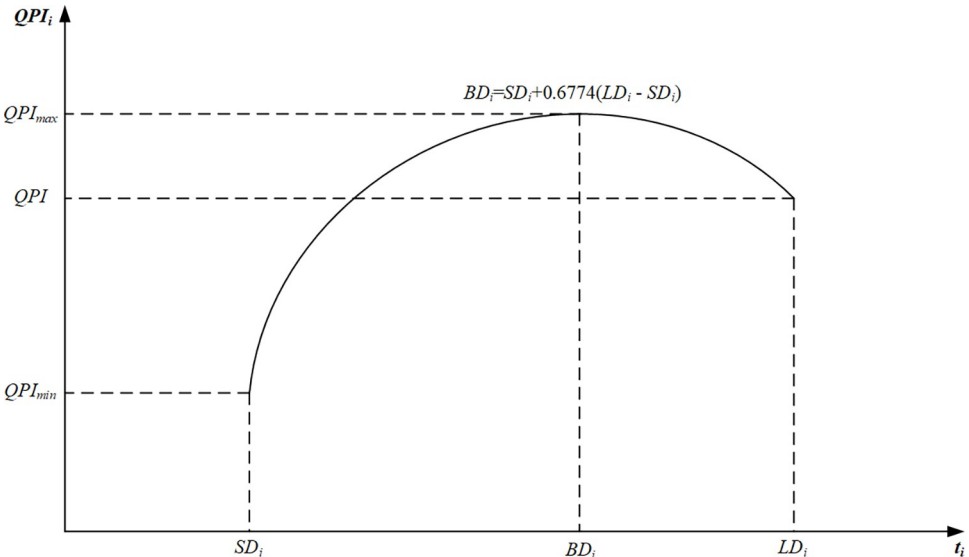

**Fig 8. The curve of the relationship of time- quality.**

Therefore, *QPI* of the whole project can be calculated as

$$QPI = \sum_{i=1}^{n} QPI_i \tag{24}$$

The maximization of total quality is shown as follow:

$$MAXQ = \sum_{i=1}^{n} QPI_i \Big/ n \tag{25}$$

## The proposed multi-objective immune wolf colony algorithm

### Principle of WCA

Wolf colony algorithm (WCA) is an intelligent optimization algorithm that simulates natural wolf hunting behaviour [40]. It can show three kinds of intelligent behaviors (wandering behavior, calling behavior and siege behavior) and update mechanism of wolves. In the WCA, the best individual is named as $\alpha$ wolf, the second-best individuals are named as $\beta$ wolves and third best individuals are named as $\gamma$ wolves, respectively, and the other individuals are called as $\omega$ wolves. Wandering behavior refers to exploring the optimal position of wolves in the current area to randomly search for prey. Calling behavior refers to the $\alpha$ wolf calling the nearby $\beta$ wolves through howling, and the $\beta$ wolves rush to the $\alpha$ wolf quickly. Siege behavior refers to the wolves' encircling of prey. Through the natural principle of "survival of the fittest", the renewal mechanism of wolf pack is to selectively eliminate some wolves with poor ability and give priority to ensure that the wolves with ability can get enough food, so as to ensure the continuity and development of wolf pack. The detailed steps of WCA are as follows:

**Step 1.** Initialization of the wolf colony

Assuming that the size of the wolf colony is ND, the size of the wolf colony search space is D, and the location of the i wolf in the early generation is:

$$X_i = (x_{i1}, x_{i2}, \ldots, x_{id}, \ldots, x_{iD}) \tag{26}$$

$$x_{id} = rand(x_{id}^M - x_{id}^m) + x_{id}^m \tag{27}$$

where rand is a random number in [0,1], $x_{id}^m$ is the minimum range of iteration $x_{id}$, $x_{id}^M$ is the maximum range of iteration $x_{id}$. By calculating the objective function of wolves, comparing the value of the function, the $\alpha$ wolf, $\beta$ wolves and $\gamma$ wolves are selected from all wolf colony. The $\alpha$ wolf does not need to perform the next three intelligent behaviors, and directly inter to the next iteration until it is replaced by a better wolf or terminated.

**Step 2.** Wandering behavior of the wolf colony

After choosing the $\alpha$ wolf, the $\beta$ wolves start wandering behavior in the space. The $\beta$ wolves move in the $v$ direction and record the fitness value in each direction. Then the $\beta$ wolves will move in the direction of higher odor concentration and update the position of the $\beta$ wolves. After the $\beta$ wolf moves in the $p(p = 1,2,\ldots,v)$ direction, the position of $\beta$ wolf in the $d$ ($d = 1,2,\ldots,D$) dimension is expressed as:

$$x_{id}^p = x_{id} + \eta step_a \tag{28}$$

where $\eta$ represents the search factor of the $\beta$ wolves that is a random number in [0,1], $step_a$ represents the step of the $\beta$ wolves.

Compared the fitness value of the $\alpha$ wolf with the fitness value of the $\beta$ wolves. If the location of the wolf $x_{id}$ moves to $x_{id}^p$, the fitness of $x_{id}^p$ is better than the current location. The position of the $x_{id}$ wolf is updated to $x_{id}^p$. Otherwise, the position of the $x_{id}$ wolf does not move. The adaptation values of the $x_{id}$ wolf after updating is compared with the fitness values of the $\alpha$

wolf. If the fitness value of $\beta$ wolf better than the $\alpha$ wolf, the wolf will take the place of the $\alpha$ wolf.

**Step 3.** Calling behavior of the wolf colony

At the end of the wandering behavior, the new $\alpha$ wolf will summon the surrounding $\gamma$ wolves, the $\gamma$ wolves will run to the location of the $\alpha$ wolf with a faster speed. When wolf $i$ in the $k+1$ iteration, the position of $\gamma$ wolf in the $d(d = 1,2. . .,D)$ dimension is expressed as:

$$x_{id}^{k+1} = x_{id}^k + \theta(x_{ad}^k - x_{id}^k)step_b \tag{29}$$

Where $\theta$ represents the rush factor of the $\gamma$ wolves that is a random number in $[-1,1]$, $step_b$ represents the step of the $\gamma$ wolves, $x_{id}^k$ represents the position of the wolf $i$ in the $d$ dimensional space when the $k$ generation, $x_{ad}^k$ represents the position of the $\alpha$ wolf in the $d$ dimensional space when the $k$ generation wolves.

**Step 4.** Siege behavior of the wolf colony

When the wolf colony finish calling behavior, the $\alpha$ wolf command the $\beta$ wolves, the $\gamma$ wolves and the $\omega$ wolves to encircle the prey with the position of the $\alpha$ wolf. For the $k+1$ generation of wolves, the siege behavior of the wolves is expressed as:

$$x_{id}^{k+1} = x_{ad}^k + \sigma step_c \tag{30}$$

Where $\sigma$ represents the siege factor of the wolves that is a random number in $[-1,1]$, $step_c$ represents the step of the $\omega$ wolves.

When the siege behavior is finished, if the function value of the position of the wolf $i$ is better than the original function value, the position of the wolf will be updated; otherwise, the position of the wolf will remain unchanged. The adaptation values of the $x_{ad}^k$ wolf after updating are compared with the fitness values of the $\alpha$ wolf, $\beta$ wolves, and $\gamma$ wolves. The best three wolves were selected as the $\alpha$ wolf, $\beta$ wolves, and $\gamma$ wolves to complete the update of the $\alpha$ wolf, $\beta$ wolves, and $\gamma$ wolves.

## Multi-objective immune wolf colony algorithm

**Design idea of the IWCA.** The immune wolf colony hybrid algorithm proposed in this paper focuses on the combination of wolf colony algorithm and immune algorithm, which makes the hybrid algorithm have the advantages of immune algorithm's excellent solving performance in combinatorial optimization and wolf colony algorithm's fast convergence in solving problems, and avoids the disadvantages of immune algorithm's slow convergence speed in optimization problems and wolf colony algorithm's easy falling into local extremum. Therefore, the immune wolf colony hybrid algorithm provides a new way to solve the time-cost quality trade-off (TCQT) problem.

When using the immune wolf colony hybrid algorithm to solve the TCQT problem, this paper takes the individual wolf colony as the antibody of the immune algorithm, the odor concentration of prey as the antigen of the immune algorithm, and the odor concentration of the individual wolf colony as the fitness value of the current solution. The process of wolf colony searching, and trapping prey is to use immune wolf colony hybrid algorithm to solve the TCQT problem iteratively. In the process of wolf pack updating, it is always hoped that the wolf with high adaptability will be left behind. However, if the superior wolf is too concentrated, it is difficult to ensure the diversity of the whole wolf colony. Therefore, by using the mechanism of antibody concentration inhibition, the antibody with low affinity and high concentration will be suppressed, and the antibody with high affinity and low concentration will be retained and promote the production, to ensure the diversity of antibody groups.

The basic principle of IWCA is to combine the immune principle of antibody concentration inhibition mechanism and immune memory function in the immune algorithm with the wolf colony algorithm, calculate the concentration of wolves(antibody) and compare it with the initial wolf concentration ($C_u$). If it is higher than $C_u$, update the wolves using IWCA, otherwise, update the wolves using the self-adapting WCA. Self-adapting WCA can accelerate search efficiency. The immune operation increases wolf colony diversity, ensures convergence speed, and improves the global search ability and accuracy.

**Steps of the IWCA.** The detailed steps of IWCA are as follows:

*Step 1.* Recognize antigen. The antigen corresponds to the problem to be optimized. Prior knowledge indicates that one should use experience, knowledge, and understanding of the problems to obtain an initial feasible region or initial feasible solutions. Define fitness function $f_u$ given by Eq (3), Eq (21), and Eq (25). Then, set the parameters of IWCA: dimension $D$ of a feasible solution, which equals the number of activities, the population size of wolf colony $ND$, the expected reproduction index $\mu$, the maximum number of iterations $T$, et al.

*Step 2.* According to Eq (27), generate the initial generation population of wolves randomly, and in the meantime record the location of each wolf.

*Step 3.* Evaluate the fitness by calculating the fitness of each current wolf (antibody). The affinity between antibodies and antigens is used to indicate the recognition degree of antibody to antigen, which is called its fitness. The affinity function is expressed as:

$$P_u = \frac{1}{f_u} \tag{31}$$

where $f_u$ is the fitness function. Then, select $\alpha$ wolf, $\beta$ wolves, $\gamma$ wolves and $\omega$ wolves by comparing the fitness value.

*Step 4.* Judge whether the terminal condition is met. If the current wolf colony includes the best wolf(antibody), which means that the antibody has the maximal affinity degree with the antigen, then output the antibody and stop searching; otherwise, go to the next step and continue searching.

*Step 5.* Calculate the antibody affinity of each wolf. The affinity between antibodies indicates the degree of similarity between antibodies. In this paper, it is measured by Forrest's R consecutive matching method. The expression of the affinity function between antibodies can be calculated as:

$$R_{u,v} = \frac{\omega_{u,v}}{L} \tag{32}$$

where $\omega_{u,v}$ represents the number of identical bits between antibody $u$ and antibody $v$. $L$ represents the length of antibody.

*Step 6.* Antibody concentration is the ratio of similar antibodies to all antibody groups. The concentration of the current antibody is expressed as:

$$C_u = \frac{\sum_{u,v \in N} R_{u,v}}{N} \tag{33}$$

$$R_{u,v} = \begin{cases} 1 & R_{u,v} > a \\ 0 & othor \end{cases} \tag{34}$$

where $C_u$ represents the concentration of antibody. $N$ represents the size of the antibody population. $R_{u,v}$ represents bivariate function, $R_{u,v}$ is 1, when the affinity between antibodies is

greater than $\alpha$, which represents the threshold value that decides whether antibody $u$ is similar to antibody $v$, otherwise $R_{u,v}$ is 0.

*Step 7*. Calculate the expected reproduction rate of the antibody population. The expected reproduction rate is composed of the affinity between antibody and antigen and antibody concentration. Expected reproduction is expressed as:

$$Y = \lambda \frac{P_u}{\sum P_u} + (1 - \lambda) \frac{C_u}{\sum C_u} \tag{35}$$

where $Y$ is the expected reproduction rate of the antibody $u$, $\lambda$ is a constant.

*Step 8*. Perform a judgment of concentration. Expected reproduction($Y$) is inversely proportional to antibody concentration ($C_u$). Compared $Y$ with $\mu$. If $Y > \mu$, indicates that $C_u$ is low, update the population using self-adapting WCA, and continue with step 9(a); if $Y \leq \mu$, indicates that $C_u$ is high, and diversity may be increased to avoid prematurity, and continue with step 9 (b).

Step 9.

a. Perform dynamic self-adapting of calling behavior of the wolf colony. For solving the local optimal solution of WCA, by referring to the adaptive parameter adjustment method in the particle swarm optimization algorithm, change the rush factor $\theta$ in Eq (29) into a dynamic weight coefficient $\theta^*$, to change the fixed step of the calling behavior. In this paper, dynamic adaptive calling behavior of WCA algorithm is proposed. The new calling behavior of the wolf colony according to the following equations:

$$x_{id}^{k+1} = x_{id}^k + \theta^*(x_{ad}^k - x_{id}^k)step_b \tag{36}$$

$$\theta^* = \begin{cases} \theta_{min}^* + \dfrac{(\theta_{max} - \theta_{min})(F_{avg}^k - F_{min}^k)}{F_{id}^k - F_{min}^k} & F_{id}^k \geq F_{avg}^k \\ \theta_{max}^* & F_{id}^k < F_{avg}^k \end{cases} \tag{37}$$

where $\theta^*$ is dynamic adaptive coefficient, $\theta_{max}^*$ and $\theta_{min}^*$ are the maximum and minimum inertial coefficients respectively; $F_{min}^k$ is the minimum value of fitness function of wolf colony in the $k$ generation; $F_{id}^k$ is the value of fitness function of the wolf $i$ in the $d$ dimensional space when the $k$ generation; $F_{avg}^k$ is the average value of fitness function.

b. Perform the immune operations of selection, crossover, and mutation. Compared with the traditional immune operation. The selection operation is a roulette wheel. However, the crossover operation and mutation operation are improved.

In this paper, a single point crossover operation is based on the same gene number. The difference between the proposed method and the classical single point crossover operation is that the selection of crossover points is not random. The specific method is to randomly select two chromosomes, select the same gene number from the two parent chromosomes as the intersection point, and then split and exchange the remaining chromosomes on the right side of this point to obtain new offspring. The new crossover operation method can improve the running speed of the algorithm and ensure the diversity of the population. A single point crossover operation based on the same gene number is shown in Fig 9.

In this paper, the mutation operation of two-way search is proposed, aim is to avoid the degradation of the antibody population. The concrete way is to randomly select M point (red square) as a new variation point. First, M point as the starting point to search, N point for the search, eventually point to reverse after the search will be able to get a new mutation individual 1. Secondly, to H point starting point for searching, M point for variation at the end of the

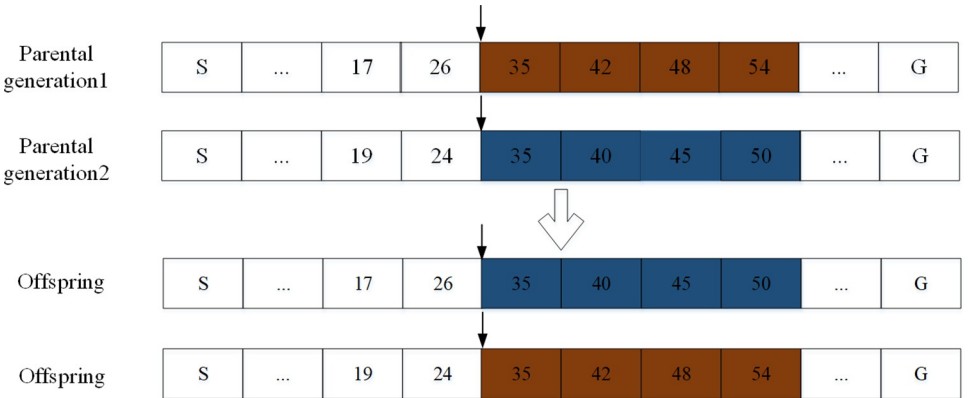

**Fig 9. Single point crossover operation based on the same gene number.**

reverse search to get a new mutation individual 2. The mutation operation based on two-way search is shown in Fig 10.

*Step 10.* Update the wolf colony. For each wolf that has been updated, compare its current fitness value with the fitness value of original best wolf ($\alpha$ wolf). If the current fitness value is better than $\alpha$ wolf, then replace $\alpha$ wolf, otherwise, retain α wolf and continue to go to step 3.

*Step 11.* Judge whether the termination conditions of the algorithm is met. If the termination conditions are met, the algorithm is ended. Otherwise, go to step 3, until the maximum number of iterations t is reached.

According to the principles described above, a flowchart of the solving process is shown in Fig 11.

## Case study

In the above content, based on grey wolf optimizer (GWO) algorithm [41] and immune mechanism, we proposed immune wolf colony algorithm (IWCA). It can find the best and unique solution when dealing with single objective problems, but IWCA cannot solve multi-objective optimization problems. The reason is due to the sub objectives of the multi-objective optimization problem are contradictory, and the improvement of one sub objective may cause the performance of another or several other sub objectives to change accordingly (such as the TCQP problem to be solved in this paper). That is, it is impossible to achieve the optimal value of

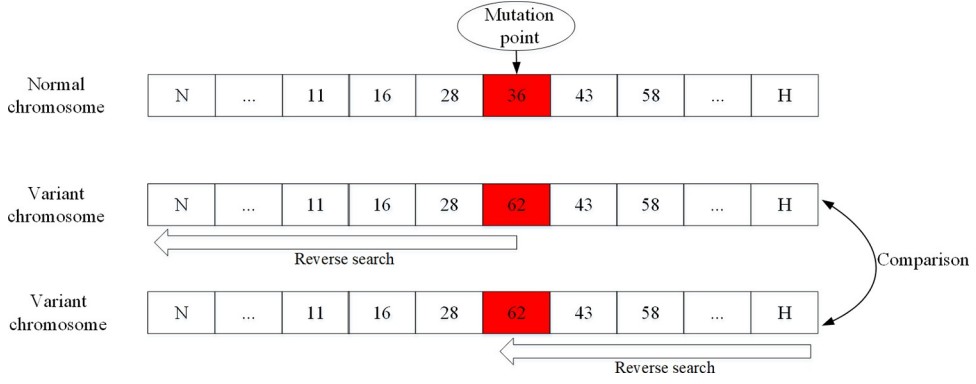

**Fig 10. Mutation operation based on two-way search.**

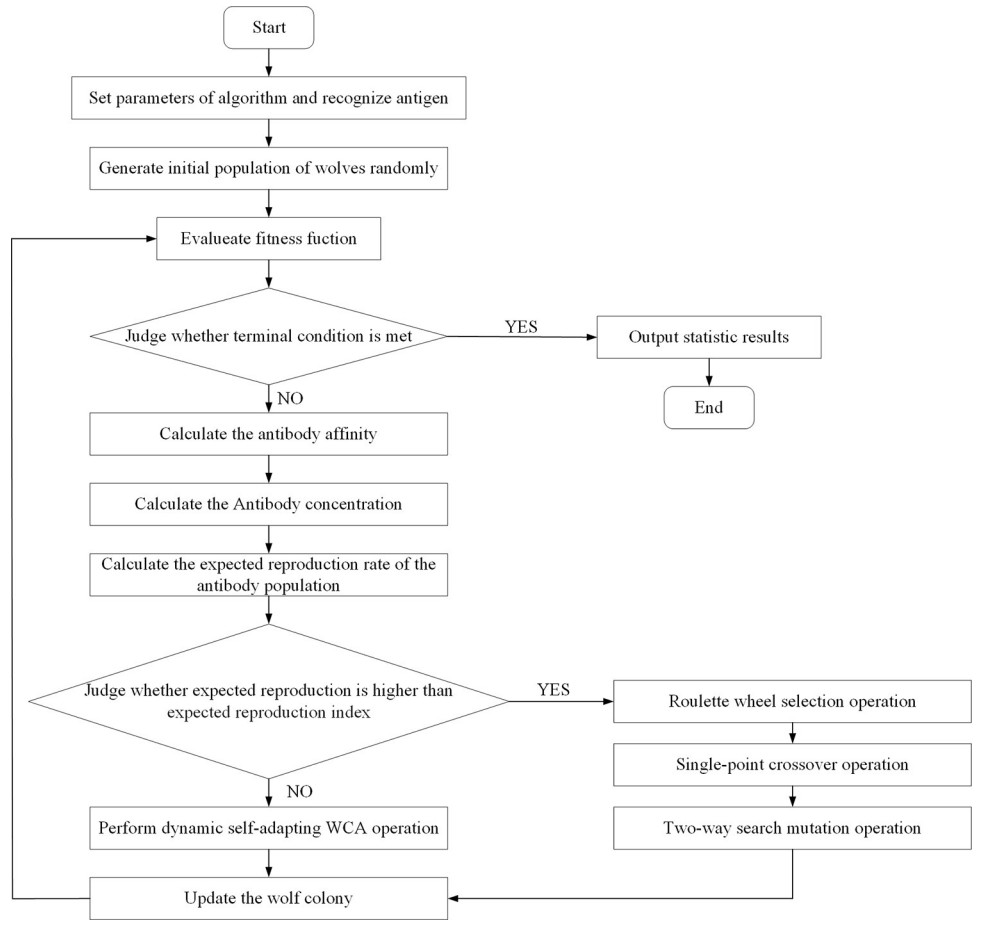

**Fig 11. Flowchart of immune wolf colony algorithm.**

multiple sub objectives at the same time, but only to coordinate and compromise among them, make each sub goal as optimal as possible. Its essential difference from the single objective optimization problem is that the solutions of the multi-objective optimization problem are not unique, but there is a set of optimal solutions composed of many Pareto optimal solutions. Each element in the set is called Pareto optimal solution or non-inferior optimal solution. In order to preform multi-objective optimization problems by IWCA, we combine the idea of multi-objective optimization with the excellent strategy of IWCA, and propose a multi-objective immune wolf colony algorithm (MOIWCA). We learn from the components used in MOPSO [42] and MOGWO [43], and integrate two new components on the basis of IWCA. The first component is to create an archive, which is responsible for storing all non-dominated Pareto optimal solutions generated during algorithm iteration. When the optimal solution is to be archived or the archive is full, it can be controlled through the archive controller. The second component is the leader selection strategy, which is used to select from files $\alpha$ wolf, $\beta$ wolves, and $\gamma$ Wolves is the leader in the hunting process. This component can select the least crowded part of the search space in the best archive, and provide one of the non-dominant solutions. The selection is done by roulette-roulette method. These two components have been described in detail in the literature [43], and we will not repeat in the paper.

## Test of multi-objective immune wolf colony algorithm

We chose the Rosenbrock function and the Rastrigin function to prove the robustness and effectiveness of MOIWCA. To prove the advantages of MOIWCA, we conducted comparative experiments with IGPSO [19], NSGA-II [44], and MOEA/D [45]. We retain the experimental conditions unchanged: the four optimization algorithms solve the two functions 10000 times each; the terminal condition of the algorithm iteration is that the number of iterations is no more than 10000, or the current global solution error is less than 0.9. Compared with the initial error value, the global solution error of 0.9 is relatively small. It is a recognized low error level and is an effective test of the convergence of the algorithm. The two test functions are as follows:

(1) The Rosenbrock functions

$$f(\mathbf{x}) = \sum_{i=1}^{N-1}[(1-x_i)^2 + 100(x_{i+1} - x_i^2)^2] \quad \forall \mathbf{x} \in \mathbb{R} \tag{38}$$

where $x = (x_1, x_2,\ldots,x_n)$ is a n-dimensional vector, with $|x_i|<2.4$ and $i = 1,2,\ldots,n$. The Rosenbrock function is a non-convex function with a global minimum. Assuming $N = 3$, the global minimum of the Rosenbrock function is 0 at (1,1,1). The location is in a parabolic valley. Because the value in the valley changes little, it is more difficult to converge to the optimal region than to the global minimum point. Therefore, this function can verify the convergence ability of MOIWCA. We set the feasible region is extended to [−100,100] to test the convergence efficiency. Table 1 shows the statistical results of the four algorithms. Compared with IGPSO, NSGA-II and MOEA/D, IWCA has a shorter running time and can find a better minimum value with a high success rate. The results reflect the effectiveness and stability of MOIWCA.

(2) The Rastrigin function

$$\boldsymbol{Ras}(x) = 20 + x_1^2 + x_2^2 - 10(\cos 2\pi x_1 + \cos 2\pi x_2) \tag{39}$$

where $x_i\in$[−5.12,5.12], the minimal point of the function is $x_i = 0$ and the global minimum is 0. The Rastrigin function has many sinusoidal and convex local minima in the solution space, which makes it more difficult for the optimization algorithm to find the global minimum. Therefore, this function can verify the convergence ability of IWCA. We expand the original feasible region 100 times, which is [−512,512], to search. The test results are listed in Table 2.

By analyzing the test results of six convergence efficiency indexes of two tests, it can be concluded that both MOIWCA, IGPSO, NSGA-II, and MOEA/D can successfully achieve the convergence conditions and find the optimal value. However, compared with the other three algorithms, MOIWCA has a shorter average cycle time, smaller average error, smaller standard deviation, and a smaller best minimum optimal value. Therefore, MOIWCA has better

**Table 1. Comparison of test results of Rosenbrock function using MOIWCA, IGPSO, NSGA-II and MOEA/D.**

|  | Average running time (ms) | Average iteration generations | Mean error | Standard deviation | Success rate (%) | Best minimum value |
|---|---|---|---|---|---|---|
| IGPSO | 8,687 | 2,183 | 0.429618 | 0.256765 | 100.00 | 0.007728 |
| MOIWCA | 7,568 | 1,750 | 0.415738 | 0.235836 | 100.00 | 0.004258 |
| NSGA-II | 15,366 | 3,628 | 0.820524 | 0.446531 | 100.00 | 0.008795 |
| MOEA/D | 11,082 | 2,825 | 0.646358 | 0.315873 | 100.00 | 0.007936 |

**Table 2. Comparison of test results of Rastrigin function using MOIWCA, IGPSO, NSGA-II and MOEA/D.**

|  | Average running time (ms) | Average iteration generations | Mean error | Standard deviation | Success rate (%) | Best minimum value |
|---|---|---|---|---|---|---|
| IGPSO | 18,046 | 3,928 | 0.367794 | 0.241865 | 100.00 | 0.004157 |
| MOIWCA | 9,658 | 2,486 | 0.285682 | 0.224783 | 100.00 | 0.003265 |
| NSGA-II | 27,863 | 6,838 | 0.824563 | 0.468735 | 100.00 | 0.013648 |
| MOEA/D | 22,648 | 5,082 | 0.693425 | 0.336749 | 100.00 | 0.008543 |

convergence efficiency and optimization ability. In the next section, MOIWCA will be tested on how excellent it can solve a real-world multi-objective optimization problem.

## Case study of a railway construction project

To prove the effectiveness of the proposed multi-objective immune wolf colony algorithm (MOIWCA) to solve the TCQT problem. This paper takes the Beijing-Shanghai high-speed railway construction project as an example. The railway construction project has 14 activities, which include construction preparation, beam making, beam erection, tunnel engineering, subgrade engineering, yellow River Bridge, ballast less track, track laying, ballast paving, ballasted track, zaobeng pilot section, communication/signal/power/electrification engineering, station house, joint commissioning. Table 3 illustrates the case study data, including the logical relationship between activities, duration, cost, and quality.

To apply the MOICWA, the algorithm parameters need to be set. The main algorithm parameters include population size $ND$, number of generations $T$, crossover probability $P_c$, mutation probability $P_m$, expected reproduction index $\mu$ and so on. Due to more thorough sampling of the state space, a larger population size may increase the probability of success in searching for optimal solution. However, more particles require more calculations, resulting in longer calculation time. Therefore, a medium-sized population is reasonable. In this study, the

**Table 3. Construction activities and corresponding parameters of Beijing-Shanghai high speed railway.**

| Activity number | Activity name | Logical | Option1 | | | Option2 | | | Option3 | | |
|---|---|---|---|---|---|---|---|---|---|---|---|
| | | | T[a] | C[b] | Q[c] | T | C | Q | T | C | Q |
| 1 | Construction preparation | — | — | — | — | 6 | 13 | 97.64 | 7 | 11 | 93.28 |
| 2 | Beam making | 1SS+2months | 20 | 230 | 85.63 | 21 | 224 | 98.45 | 22 | 220 | 92.68 |
| 3 | Beam erection | 1FS+3months | 19 | 184 | 85.39 | 20 | 176 | 98.68 | 21 | 172 | 91.76 |
| 4 | Tunnel engineering | 2 | 21 | 192 | 84.92 | 22 | 185 | 98.11 | 23 | 181 | 91.75 |
| 5 | Subgrade engineering | 2 | 27 | 263 | 86.28 | 28 | 257 | 99.34 | 29 | 252 | 93.26 |
| 6 | Yellow River Bridge | 5SS-1month | 26 | 242 | 83.87 | 27 | 238 | 98.69 | 28 | 235 | 92.43 |
| 7 | Ballastless track | 2FS-4months | 12 | 130 | 82.48 | 13 | 125 | 96.58 | 14 | 123 | 91.35 |
| 8 | Track laying | 5FS-1month | 5 | 18 | 81.74 | 6 | 16 | 95.69 | 7 | 13 | 91.26 |
| 9 | Ballast paving | 8 | 4 | 15 | 82.35 | 5 | 13 | 96.71 | 6 | 11 | 91.67 |
| 10 | Ballasted track | 9 | 3 | 12 | 82.63 | 4 | 10 | 96.95 | 5 | 9 | 90.89 |
| 11 | Zaobeng pilot section | 1 | 32 | 313 | 84.16 | 33 | 306 | 98.82 | 34 | 302 | 93.65 |
| 12 | Communication/signal/power/electrification engineering | 5FS | 7 | 23 | 83.73 | 8 | 20 | 97.13 | 9 | 18 | 93.11 |
| 13 | Station house | 3SS+5months | 24 | 164 | 83.52 | 25 | 158 | 95.34 | 26 | 155 | 92.78 |
| 14 | Joint commissioning | 13FS | — | — | — | 3 | 5 | 99.86 | — | — | — |

[a]Project time (months)

[b]Project cost (hundred-million-yuan)

[c]Project quality (%).

**Table 4. Best MOIWCA parameter for 14 activities in the Beijing-Shanghai high speed railway project.**

| Parameters | Significance | Value |
|---|---|---|
| $ND$ | Population size | 112 |
| $T$ | Number of generations | 300 |
| $P_c$ | Crossover probability | 0.85 |
| $P_m$ | Mutation probability | 0.055 |
| $\mu$ | Expected reproduction index | 0.95 |
| $step_a$ | Wandering step | 0.9 |
| $step_b$ | Calling step | 0.6 |
| $step_c$ | Siege step | 0.5 |
| $\lambda$ | Constant of expected reproduction | 0.3 |

value of 8 times the number of activities is considered as the population size, so $ND = 8^*14 = 112$, other parameter values are shown in Table 4.

Because there are 3 options of construction methods for each of the 14 activities, the total number of the feasible method-combinations is up to $3^{14} = 4782969$. Each possible combination will have a unique impact on the project objectives, such as the shortest time, minimum cost, and maximum quality of the project. The multi-objective optimization model proposed in this paper is to search the large search space of possible solutions. By finding the non-dominated solution in successive generations, this method can narrow down the large search space.

Table 5 shows the best solution and project objectives. Solution 13 generated the minimum duration of the project, solution 12 generated the minimum cost of the project, and solution 16 generated the highest quality of the project. In addition to these three solutions, the remaining solutions represent the trade-off between time, cost, and quality.

The project manager can choose the best solution for a specific project scenario according to decisions, preferences, and conditions. For example, if the project manager gives priority to the project with the minimum time, then solution 13 should be selected to achieve the minimum time value of 214 months by increasing additional project cost with low project quality. If the project manager gives priority to the minimum cost of the project, then solution 12 should be selected to achieve the minimum cost value of 1729 hundred million yuan by increasing additional project time with high project quality. If the project manager gives priority to the project with the highest quality, then solution 16 should be selected to achieve the maximum quality value of 95.84%. In addition, if the project manager wants to strike a measured balance between these three objectives, then solution 10 provides the compromise solution, including acceptable project time (219 months), project cost (1764 hundred million yuan) and project quality (92.99%). Fig 12 shows the typical Pareto optimal fronts obtained using the MOIWCA for this case study. These three fronts clearly show the relationships among project duration, cost, and quality. It is not difficult to see that the relationship between time, cost and quality is non-linear. For example, the project manager can appropriately extend the time and reduce the cost according to the actual situation of the construction project, and the quality can meet the requirements of the construction project; Similarly, the quality can also meet the requirements of the construction project by appropriately reducing the time but increasing the cost. However, the quality of the construction project cannot meet the minimum requirements if the cost is constantly increased by blindly seeking to reduce the time of the construction project or blindly increasing the time to reduce the cost. This three-dimensional representation of the tradeoffs may help decision-makers figure out how the different possible plans for using resources will affect the performance of the project.

**Table 5. Best non-dominated solutions obtained by MOIWCA-TCQT.**

| Solution | Execution method combination | Project Performance | | |
|---|---|---|---|---|
| | | Time (months) | Cost (hundred million yuan) | Quality (%) |
| 1 | [2, 2, 1, 1, 2, 2, 2, 2, 3, 2, 2, 1, 1, 2] | 218 | 1768 | 93.66 |
| 2 | [3, 2, 2, 2, 1, 3, 1, 2, 2, 1, 3, 2, 1, 2] | 220 | 1756 | 92.79 |
| 3 | [2, 3, 2, 1, 1, 1, 3, 2, 2, 3, 1, 1, 2, 2] | 219 | 1766 | 91.56 |
| 4 | [2, 2, 2, 2, 1, 1, 1, 2, 3, 2, 1, 2, 2, 2] | 218 | 1766 | 93.36 |
| 5 | [2, 3, 3, 1, 1, 1, 2, 2, 3, 1, 1, 1, 2, 2] | 218 | 1765 | 90.49 |
| 6 | [3, 2, 1, 3, 3, 3, 2, 3, 1, 1, 1, 1, 2, 2] | 221 | 1751 | 90.76 |
| 7 | [2, 1, 1, 3, 1, 3, 2, 1, 2, 1, 3, 1, 1, 2] | 217 | 1768 | 89.82 |
| 8 | [3, 1, 1, 2, 3, 1, 2, 3, 1, 1, 2, 3, 3, 2] | 221 | 1753 | 91.21 |
| 9 | [2, 1, 1, 1, 2, 2, 1, 3, 2, 1, 2, 2, 3, 2] | 218 | 1768 | 92.38 |
| 10 | [2, 2, 1, 3, 2, 2, 2, 1, 2, 3, 1, 2, 1, 2]<sup>d</sup> | 219 | 1764 | 92.99 |
| 11 | [3, 1, 1, 2, 2, 3, 2, 2, 1, 2, 3, 3, 2, 2] | 222 | 1751 | 93.41 |
| 12 | [3, 3, 2, 2, 2, 3, 3, 1, 3, 3, 2, 3, 3, 2]<sup>b</sup> | 228 | 1729 | 93.91 |
| 13 | [2, 2, 1, 1, 1, 1, 1, 2, 1, 1, 2, 1, 3, 2]<sup>a</sup> | 214 | 1780 | 89.64 |
| 14 | [2, 3, 3, 1, 2, 2, 1, 1, 1, 3, 2, 1, 2, 2] | 219 | 1756 | 91.45 |
| 15 | [2, 3, 1, 2, 3, 1, 2, 3, 1, 1, 2, 1, 2, 2] | 219 | 1753 | 91.54 |
| 16 | [2, 3, 2, 2, 3, 2, 2, 2, 2, 1, 2, 2, 2, 2]<sup>c</sup> | 222 | 1739 | 95.84 |
| 17 | [3, 1, 1, 1, 2, 2, 2, 3, 1, 3, 2, 1, 1, 2] | 218 | 1772 | 91.02 |
| 18 | [3, 2, 2, 1, 2, 1, 1, 2, 2, 3, 1, 1, 3, 2] | 219 | 1766 | 91.77 |
| 19 | [2, 1, 1, 1, 1, 1, 3, 3, 3, 1, 1, 1, 2, 2] | 216 | 1782 | 88.84 |
| 20 | [3, 2, 3, 1, 1, 1, 2, 2, 3, 1, 1, 3, 1, 2] | 219 | 1768 | 90.41 |

[a] minimum time

[b] minimum cost

[c] maximum quality

[d] best compromise.

The non-dominated solutions may also be used to optimize tradeoffs between any two objectives on a two-dimensional plane. Figs 13–15 show the trade-off between time and cost, time and quality, and cost and quality, respectively. As illustrated in Fig 13, if the investment in the construction project is increased, the duration of completing the project will be shortened, and vice versa. As shown in the Fig 14, if the duration of the construction project is increased, the quality of the completed project will increase first and then decrease. The situation shows that excessive extension of the construction project will lead to a decline in the quality of the project. One of the possible reasons for the situation is that long hours of work will reduce the working efficiency of labors. As shown in the Fig 15, if the quality of construction projects is improved, the investment in construction projects will increase. However, blindly increasing investment will not make the quality higher and higher but can reduce the quality. One of the possible reasons for the situation is that the purpose of increasing costs is to shorten the duration of the construction project, and the reduction of time leads to the decline of the quality of the construction project.

## Performance comparison and analysis

We compared MOIWCA performance against NSGA-II, MOPSO and MODE to assess comparative effectiveness. For comparison purposes, all four algorithms used an equal number of function evaluations, had a population size of 300, and a maximum of 500 generations. In NSGA-II, the crossover probability is set at 0.5 and the mutation probability is set at 0.1. In

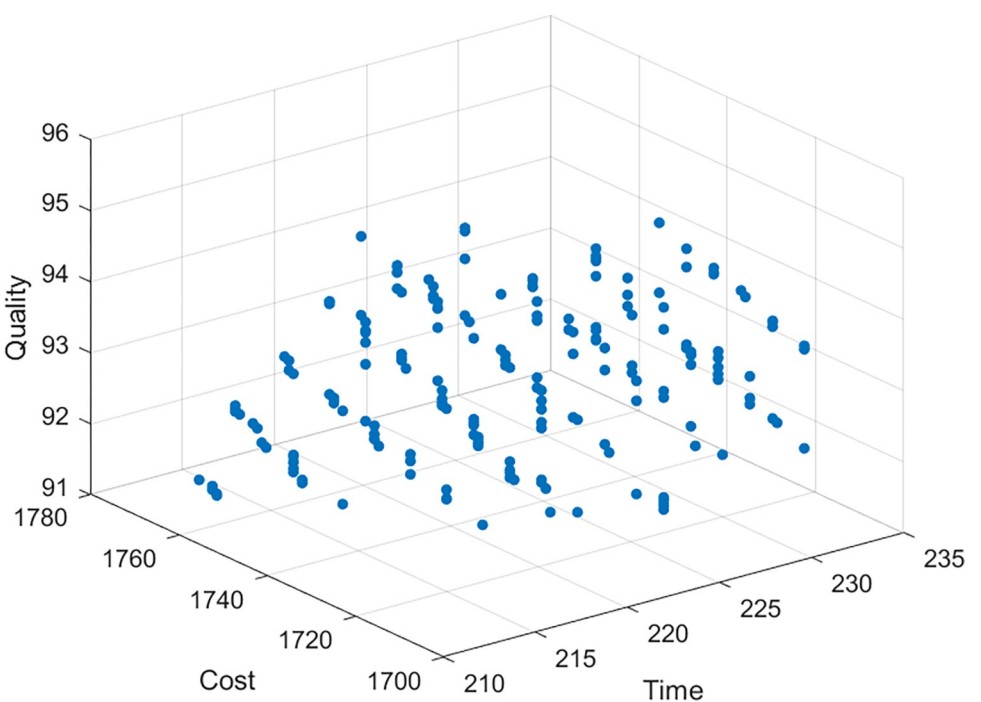

**Fig 12. Time–cost–quality tradeoff Pareto front using MOIWCA.**

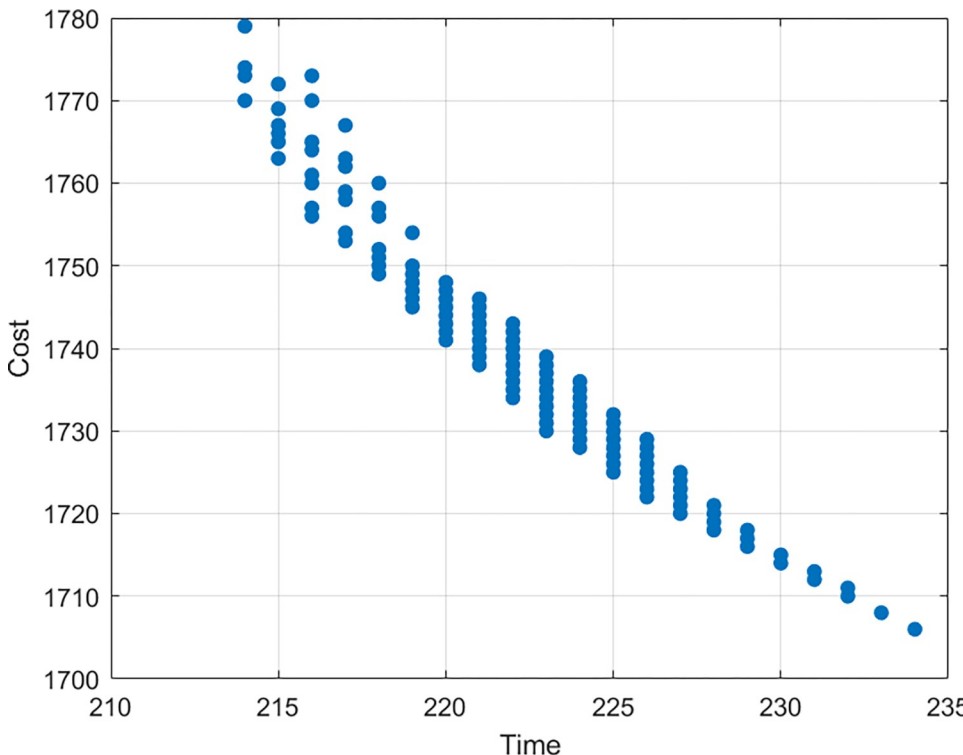

**Fig 13. Time-cost tradeoff analysis.**

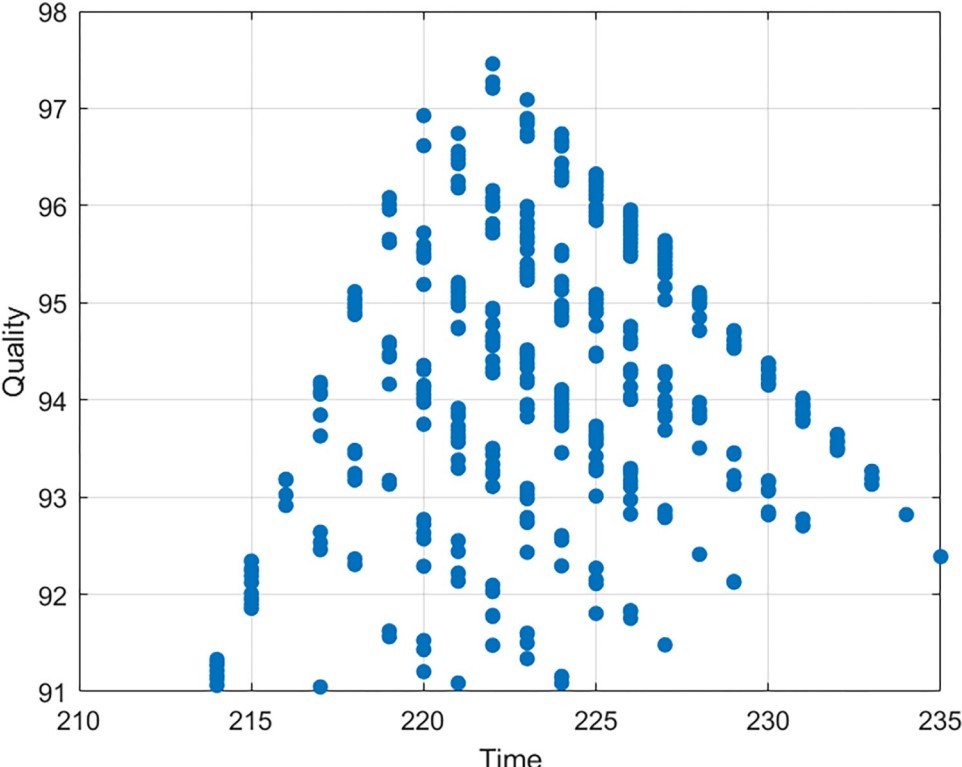

**Fig 14. Time-quality tradeoff analysis.**

MOPSO, the two learning factors $c_1$, $c_2$ are both chosen at 1.5, and the inertia factor $w$ is set in the range of 0.2–0.6. The MOIWCA control parameters remained the same, as stated previously in Table 4. Thirty independent runs are carried out for all experiments.

Multi-objective optimization problem performance measures are more complex than those of single-objective optimization problems. Three issues are typically considered: (1) convergence to the Pareto optimal set; (2) preservation of diversity in Pareto optimal set solutions; and (3) the maximal distribution bound of the Pareto optimal set [46]. In the literature, the researchers have suggested numerous quality indicators. In this paper, the Spacing metric, and the Hyper volume (HV) metric are selected to evaluate the performance of different algorithms.

(1) Spacing metric: This indicator assesses the degree of distribution of non-dominated solutions. The mathematical definition of the spacing metric may be given as:

$$\Delta = \frac{\sum_{i=1}^{m} d(E_i, \Omega) + \sum_{X \in \Omega} |d(X, \Omega) - \bar{d}|}{\sum_{i=1}^{m} d(E_i, \Omega) + (|\Omega| - m)\bar{d}} \tag{40}$$

where $\Omega$ is a set of non-dominated Pareto-front solutions, $(E_1, E_2, \ldots, E_k)$ are $k$ extreme solutions in the set is true Pareto-front, $m$ is the number of objectives and $d(X, \Omega) = \min_{Y \in \Omega, Y \neq X} \|F(X) - F(Y)\|$ is the minimum Euclidean distance between solution X and its neighboring solutions in the obtained non-dominated X set, $\bar{d} = \frac{1}{|\Omega|} \sum_{X \in \Omega} d(X, \Omega)$ is the mean value of all $d(X, \Omega)$, $|\Omega|$ is the total solutions in $\Omega$ set. A value of 0 indicates that all members of Pareto optimal solution are equally spaced. Table 6 shows the comparison of the Spacing metric for

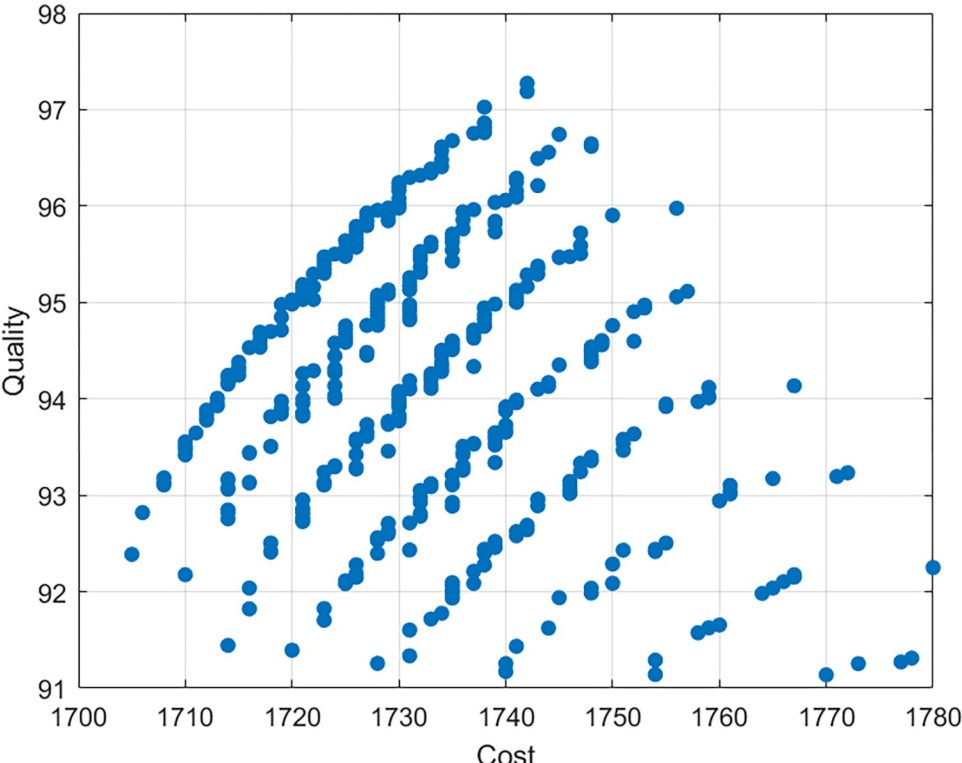

**Fig 15. Cost-quality tradeoff analysis.**

different algorithms. Fig 16 is the visualization of the results. The smaller the value of the Spacing metric, the better distribution and diversity of non-dominated solutions. From the comparison results of the Spacing metric for different algorithms, the MOIWCA obtained the minimum Spacing metric. This shows that in dealing with multi-objective optimization problems, The non-dominated solutions obtained by MOIWCA have better distribution and diversity than other algorithms.

(2) Hyper-volume (HV): This indicator calculates the volume (in the objective space) covered by members of a non-dominated set of solutions $\Omega$ to a problem that works to minimize all objectives. A hypercube $v_i$ is constructed for each solution $X_i \in \Omega$, with reference point $W$ and the solution $X_i$ as the diagonal corners of the hypercube. The reference point may be found simply by constructing a vector of the worst objective function values. Thereafter, a union of all hypercubes is found, with the HV of this union calculated as:

$$HV = \bigcup_{i=1}^{|\Omega|} v_i \tag{41}$$

Algorithms with larger HV values are desirable. The HV value of a set of solutions is

**Table 6. Comparison of Spacing metric for different algorithm.**

|  | MOIWCA | MODE | MOPSO | NSGA-II |
|---|---|---|---|---|
| Best | 0.4336 | 0.4453 | 0.4627 | 0.4852 |
| Worst | 0.6132 | 0.7025 | 0.9416 | 0.9653 |
| Average | 0.5235 | 0.5431 | 0.7123 | 0.7356 |
| Standard Deviation | 0.0397 | 0.0586 | 0.1542 | 0.1653 |

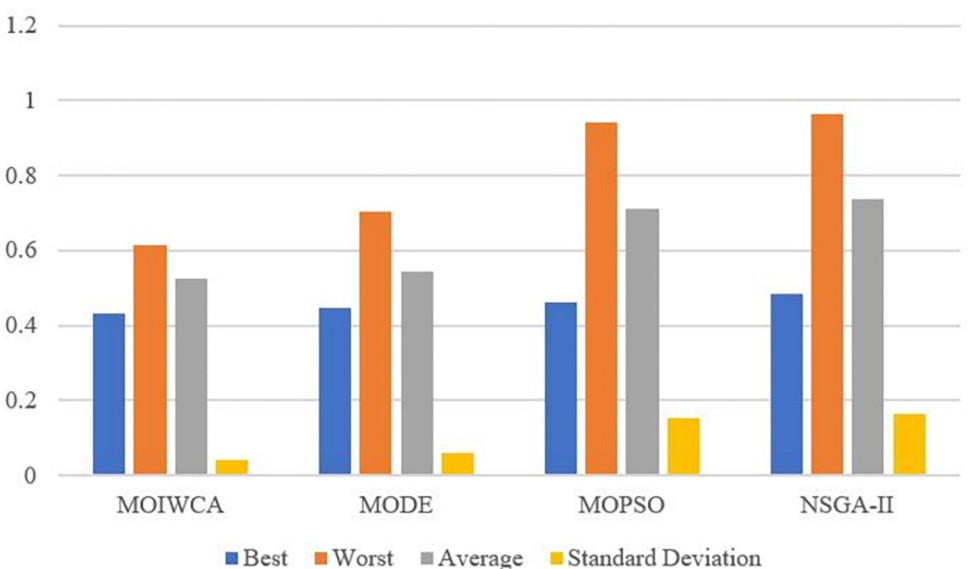

**Fig 16. Comparison of Spacing metric for different algorithm.**

**Table 7. Comparison of HV metric for different algorithm.**

|  | MOIWCA | MODE | MOPSO | NSGA-II |
|---|---|---|---|---|
| Best | 0.9486 | 0.8965 | 0.9284 | 0.8652 |
| Worst | 0.7164 | 0.6932 | 0.4683 | 0.2473 |
| Average | 0.8207 | 0.7648 | 0.7136 | 0.5241 |
| Standard Deviation | 0.0519 | 0.0546 | 0.1483 | 0.2062 |

normalized using a reference set of Pareto optimal solutions with the same reference point. After normalization, the HV values are confined to range [0,1]. Table 7 lists the results for each of the four compared algorithms in terms of HV. From Table 7, we can see that the proposed model gets the highest HV values. This means that the MOIWCA has better convergence and diversity performance than the other algorithms. Fig 17 is the visualization of the results.

## Conclusions

In this paper, the time-cost-quality trade-off problem is studied, and a hybrid algorithm MOIWCA is developed to solve the TCQT problem for construction project. The paper makes three important contributions: first, based on the traditional time-cost model, we introduce the bonus-penalty mechanism, and propose a new nonlinear time-cost model. Meanwhile, we develop a new QPI model by analyzing the qualitative and quantitative relationship between time and quality. Second, we propose a multi-objective immune wolf colony algorithm (MOIWCA)to solve the TCOQ problem. The design of algorithm is divided into two parts: we propose IWCA that combines the wolf colony algorithm and immune algorithm, and improves the crossover operation and mutation operation. In addition, we integrate IWCA with multi-objective optimization to enable it to deal with time-cost-quality trade-off problem. Thirdly, we take the Beijing-Shanghai high-speed railway construction project as a case to evaluate the performance of the MOIWCA. The results show that the MOIWCA is more effective and efficient than widely used multi-objective algorithms (MOPSO, NSGA-II, MODE). The

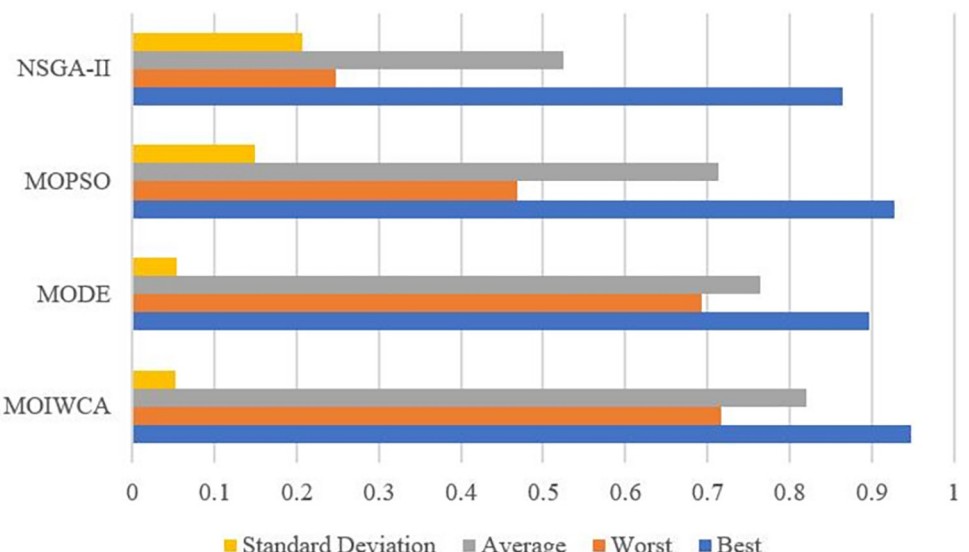

**Fig 17. Comparison of HV metric for different algorithm.**

SP and HV values comparisons indicated that the proposed MOIWCA performs excellent in terms of distribution, convergence and uniformity.

Results show that the proposed method MOIWCA generates a better Pareto front than widely used approaches. With the obtained non-dominated solutions, the project managers can easily trade-off the three important objectives, including project time, cost, and quality.

The proposed multi-objective immune wolf colony algorithm is robust and efficient. Without limitation in the setting of the number of decision variables and objectives, the MOIWCA-TCQCT model can easily apply to other multi-objective optimization problems in the field of construction projects by minor modification, such as time, cost, and risk tradeoff problems in construction management and performance, cost, and reliability in engineering design work.

Future research work will be carried out in two directions: target model and algorithm improvement. Considering the complexity of the construction project, the time-cost-quality trade-off problem cannot meet the decision-making requirements of the project manager. Problems of risk and ecology also have an important impact on construction projects. Based on the existing time-cost quality model, two objective functions of minimizing risk and maximizing ecology are added to better describe the construction project problem. But there aren't many ways to test the trade-off problem of time-cost-quality-risk-ecology or figure out how to solve it. Therefore, it is necessary to propose a new solution to deal with the problem of five objective functions. MOIWCA has excellent performance in avoiding premature convergence and improving the efficiency of initial convergence. It can further upgrade the algorithm to make it an entirely parameter-free algorithm to better solve the complex multi-objective trade-off problem.

## Supporting information

**S1 File. Design diagram of Beijing Shanghai high speed railway construction project.** (PDF)

## Author Contributions

**Data curation:** Guanyi Liu.

**Funding acquisition:** Xuemei Li.

**Investigation:** Khalid Mehmood Alam.

**Methodology:** Guanyi Liu, Xuemei Li.

**Software:** Guanyi Liu, Xuemei Li.

**Supervision:** Xuemei Li.

**Validation:** Guanyi Liu.

**Visualization:** Guanyi Liu.

**Writing – original draft:** Guanyi Liu.

**Writing – review & editing:** Xuemei Li, Khalid Mehmood Alam.

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
