## [Decision Letter · Decision Letter 0]

20 Sep 2022

PONE-D-22-22367Multiple objective immune wolf colony algorithm for solving time-cost-quality trade-off problemPLOS ONE

Dear Dr. Li,

Thank you for submitting your manuscript to PLOS ONE. After careful consideration, we feel that it has merit but does not fully meet PLOS ONE’s publication criteria as it currently stands. Therefore, we invite you to submit a revised version of the manuscript that addresses the points raised during the review process. Both reviewers have provided helpful comments, so please take this opportunity to improve your work. However, please note that they have suggested some references too. Please note that you are not required to include them in the revised draft if not relevant. Only relevant  works should be included and used to improve the literature review and comparison. ** **

We look forward to receiving your revised manuscript.

Kind regards,

Seyedali Mirjalili

Academic Editor

PLOS ONE

Journal Requirements:

Reviewers' comments:

Reviewer's Responses to Questions

**Comments to the Author**

1. Is the manuscript technically sound, and do the data support the conclusions?

Reviewer #1: Yes

Reviewer #2: Yes

2. Has the statistical analysis been performed appropriately and rigorously? 

Reviewer #1: Yes

Reviewer #2: N/A

3. Have the authors made all data underlying the findings in their manuscript fully available?

Reviewer #1: Yes

Reviewer #2: Yes

4. Is the manuscript presented in an intelligible fashion and written in standard English?

Reviewer #1: Yes

Reviewer #2: Yes

5. Review Comments to the Author

Reviewer #1: 1. In the introduction, the authors provide reviews on metaheuristics and multi-objective optimization. However, what are the drawbacks of current research? Please provide the motivation of the manuscript. Besides, what are the contributions of this manuscript?

2.The author lacks some state-of-the-art algorithms. For research on multi-objective algorithms, the compared algorithms in this manuscript are limited. Please add some state-of-the-art algorithms, e.g., NSGA-II, MOEA/D,MOSPO, MOAOS,MOAVOVA, and some multi-objective optimization methods.

3. Please use more powerful criteria such as HV or Delta. GD and IGD are basically the same.

4. Please review the paper for proper use of the English language.

5. please update ref with following:

Khodadadi, N., Azizi, M., Talatahari, S., & Sareh, P. (2021). Multi-objective crystal structure algorithm (MOCryStAl): introduction and performance evaluation. IEEE Access, 9, 117795-117812.

Azizi, M., Talatahari, S., Khodadadi, N., & Sareh, P. (2022). Multi-Objective Atomic Orbital Search (MOAOS) for Global and Engineering Design Optimization. IEEE Access.

Khodadadi, N., Abualigah, L., & Mirjalili, S. (2022). Multi-objective Stochastic Paint Optimizer (MOSPO). Neural Computing and Applications, 1-24.

Reviewer #2: This study introduces the bonus-penalty mechanism to improve the traditional time cost model, and a nonlinear time cost quality model presents to consider the nonlinear relationship between quality and time. a multi-objective immune wolf colony optimization algorithm is proposed to better solve the TCQT problem. There are the following concerns, that I highly encourage the authors to resolve them.

1. The readability and presentation of the study should be further improved. The paper suffers from language problems. It is suggested to proofread by a native speaker or a proofreading agent.

2. The importance of this study is not clear. The authors are suggested to state the reason for proposing this algorithm.

3. It is recommended to revise the contribution of this study by highlighting its novelty and finding of this study.

4. This study suffers from a lack of deep literature review on new and improved metaheuristic algorithms such as Migration-based moth-flame optimization algorithm, Hybridizing of whale and moth-flame optimization algorithms to solve diverse scales of optimal power flow problem, An improved moth-flame optimization algorithm with adaptation mechanism to solve numerical and mechanical engineering problems, and GGWO: Gaze cues learning-based grey wolf optimizer and its applications for solving engineering problems.

5. It is recommended to consider the flowchart or pseudocode of the proposed algorithm.

6. It is recommended to experimentally report the weak points of the immune wolf colony algorithm. then clarify how the proposed algorithm can overcome these weak points to solve the time-cost-quality trade-off problem.

7. Equations 36 and 37 are not clear. Please clarify them.

8. It is recommended to boost the visualization of this study by increasing the experimental evaluations.

9. It is recommended to determine the claims of this study and which experimental evaluation can support them.

10. The discussion section should be added in a more highlighting, argumentative way. The author should analyze the reason why the tested results are achieved.

11. The single-point crossover operation based on the crossover of points with the same serial number is not clear. Please increase the understanding and impact of this operator in the proposed algorithm.

12. The statistical evaluation analysis is recommended.

13. It is recommended to increase the understanding of Figs. 5 and 9.

14. The figures before equations 7-10 are not clear.

15. The nomenclature table is recommended for the parameters used in this study.

6. PLOS authors have the option to publish the peer review history of their article (what does this mean?). If published, this will include your full peer review and any attached files.

Reviewer #1: **Yes: **Nima Khodadadi

Reviewer #2: No

---

## [Author Response · Author response to Decision Letter 0]

26 Oct 2022

Reviewer #1:

Comment to the Authors

 In the introduction, the authors provide reviews on metaheuristics and multi-objective optimization. However, what are the drawbacks of current research? Please provide the motivation of the manuscript. Besides, what are the contributions of this manuscript?

Response

Thank you for your suggestion. As suggested by reviewer, we have added the suggested content to the manuscript (Page7, lines 131-145). The details are as follows:

It is not difficult to conclude from the above literature review that TCQT has been widely addressed and effectively resolved. However, the existing TCQT mathematical model has not considered the impact of reward and punishment factors on construction projects, and intelligent optimization algorithms with better solution performance need to be proposed. Thereby, the contributions of this study are stated as follows: (1) clarify the nonlinear relationship between time and bonus-penalty cost, a new bonus-penalty cost model is presented. Therefore，a new multi-objective mathematical model of the time-cost quality trade-off problem with several equality and inequality constraints is established. (2) A new multi-objective immune wolf swarm algorithm has been developed to solve the trade-off problem of time, cost, and quality. The proposed MOIWCA can solve the problem that the traditional wolf colony algorithm is easy to fall into a local optimal solution. At the same time, in order to improve the search speed and optimization performance of the algorithm, the cross operation and immune operation in the immune algorithm are improved. The algorithm can provide a representative and managed Pareto set for the time-cost-quality trade off problem. (3) To make sure that the proposed MOIWCA works and can be used, we use a high-speed railway construction project as a case study to show that the proposed method is better.

 The author lacks some state-of-the-art algorithms. For research on multi-objective algorithms, the compared algorithms in this manuscript are limited. Please add some state-of-the-art algorithms, e.g., NSGA-II, MOEA/D, MOSPO, MOAOS, MOAVOVA, and some multi-objective optimization methods.

Response

Thank you for your suggestion. As suggested by reviewer, we have added NSGA-II and MOEA/D, to further strengthen the comparison and analysis between different algorithms in the manuscript (Pages 26-27, lines 524-525 and 533-534). The details are as follows:

Table 1. Comparison of Test Results of Rosenbrock Function Using IWCA, IGPSO, NSGA-II and MOEA/D

 Average running time (ms) Average iteration generations Mean error Standard deviation Success rate (%) Best minimum value

IGPSO 8,687 2,183 0.429618 0.256765 100.00 0.007728

IWCA 7,568 1,750 0.415738 0.235836 100.00 0.004258

NSGA-II 15,366 3,628 0.820524 0.446531 100.00 0.008795

MOEA/D 11,082 2,825 0.646358 0.315873 100.00 0.007936

Table 2. Comparison of Test Results of Rastrigin Function Using IWCA, IGPSO, NSGA-II and MOEA/D

 Average running time (ms) Average iteration generations Mean error Standard deviation Success rate (%) Best minimum value

IGPSO 18,046 3,928 0.367794 0.241865 100.00 0.004157

IWCA 9,658 2,486 0.285682 0.224783 100.00 0.003265

NSGA-II 27,863 6,838 0.824563 0.468735 100.00 0.013648

MOEA/D 22,648 5,082 0.693425 0.336749 100.00 0.008543

 Please use more powerful criteria such as HV or Delta. GD and IGD are basically the same.

Response

Thank you for your suggestion. As suggested by reviewer, we select SP and HV for analysis in the manuscript (Pages 31-33, lines 614-652). The details are as follows：

We compared MOIWCA performance against NSGA-II, MOPSO and MODE to assess comparative effectiveness. For comparison purposes, all four algorithms used an equal number of function evaluations, had a population size of 300, and a maximum of 500 generations. In NSGA-II, the crossover probability is set at 0.5 and the mutation probability is set at 0.1. In MOPSO, the two learning factors c1, c2 are both chosen at 1.5, and the inertia factor w is set in the range of 0.2–0.6. The MOIWCA control parameters remained the same, as stated previously in Table 3. Thirty independent runs are carried out for all experiments. 

Multi-objective optimization problem performance measures are more complex than those of single-objective optimization problems. Three issues are typically considered: (1) convergence to the Pareto optimal set; (2) preservation of diversity in Pareto optimal set solutions; and (3) the maximal distribution bound of the Pareto optimal set [38]. In the literature, the researchers have suggested numerous quality indicators. In this paper, the Spacing metric, and the Hyper volume (HV) metric are selected to evaluate the performance of different algorithms.

(1) Spacing metric: This indicator assesses the degree of distribution of non-dominated solutions. The mathematical definition of the spacing metric may be given as:

∆=(∑_(i=1)^m▒〖d(E_i,Ω)+∑_(X∈Ω)▒|d(X,Ω)-d ® | 〗)/(∑_(i=1)^m▒d(E_i,Ω) +(|Ω|-m)d ® ) (40)

where Ω is a set of non-dominated Pareto-front solutions, (E_1,E_2,…,E_k) are k extreme solutions in the set is true Pareto-front, m is the number of objectives and d(X,Ω)=min¦(Y∈Ω,Y≠X) ‖F(X)-F(Y)‖ is the minimum Euclidean distance between solution X and its neighboring solutions in the obtained non-dominated X set, d ®=1/|Ω| ∑_(X∈Ω)▒d(X,Ω) is the mean value of all d(X,Ω), |Ω| is the total solutions in Ω set. A value of 0 indicates that all members of Pareto optimal solution are equally spaced. The smaller the value, the better distribution and diversity of non-dominated solutions. Table 6 shows a comparison of the Spacing metric for different algorithms. 

Table 6. Comparison of Spacing metric for different algorithm

 MOIWCA MODE MOPSO NSGA-II

Best 0.4336 0.4453 0.4627 0.4852

Worst 0.6132 0.7025 0.9416 0.9653

Average 0.5235 0.5431 0.7123 0.7356

Standard Deviation 0.0397 0.0586 0.1542 0.1653

(2) Hyper-volume (HV): This indicator calculates the volume (in the objective space) covered by members of a non-dominated set of solutions Ω to a problem that works to minimize all objectives. A hypercube v_iis constructed for each solution X_i∈Ω, with reference point W and the solution Xi as the diagonal corners of the hypercube. The reference point may be found simply by constructing a vector of the worst objective function values. Thereafter, a union of all hypercubes is found, with the HV of this union calculated as:

HV=⋃_(i=1)^|Ω|▒v_i (41)

Algorithms with larger HV values are desirable. The HV value of a set of solutions is normalized using a reference set of Pareto optimal solutions with the same reference point. After normalization, the HV values are confined to range [0,1]. Table 7 lists the results for each of the four compared algorithms in terms of HV. From Table 7, we can see that the proposed model gets the highest HV values. This means that the MOIWCA has better convergence and diversity performance than the other algorithms.

Table 7. Comparison of HV metric for different algorithm

 MOIWCA MODE MOPSO NSGA-II

Best 0.9486 0.8965 0.9284 0.8652

Worst 0.7164 0.6932 0.4683 0.2473

Average 0.8207 0.7648 0.7136 0.5241

Standard Deviation 0.0519 0.0546 0.1483 0.2062

 Please review the paper for proper use of the English language.

Response

Thank you for your careful review. The manuscript has been thoroughly revised and rewritten by a native English speaker.

 Please update ref with following:

Khodadadi, N., Azizi, M., Talatahari, S., & Sareh, P. (2021). Multi-objective crystal structure algorithm (MOCryStAl): introduction and performance evaluation. IEEE Access, 9, 117795-117812.

Azizi, M., Talatahari, S., Khodadadi, N., & Sareh, P. (2022). Multi-Objective Atomic Orbital Search (MOAOS) for Global and Engineering Design Optimization. IEEE Access.

Khodadadi, N., Abualigah, L., & Mirjalili, S. (2022). Multi-objective Stochastic Paint Optimizer (MOSPO). Neural Computing and Applications, 1-24.

Response

We are extremely grateful to you for pointing out this problem. The references you provided are very consistent with the research content, and have high academic value, which is very helpful to our paper. Therefore, we decided to use the three articles you suggested to further enrich our references in the manuscript (Pages 5-6, lines 113-117 and 122-130). The details are as follows：

In Khodadadi N, Azizi M, Talatahari S, Sareh P [25] considering the multi-objective optimization problem of multiple performance indicators inspired by the principle of crystal structure formation, a meta heuristic algorithm called the Crystal Structure algorithm (Crystal) is proposed, and the algorithm is evaluated. The results show that the algorithm provides excellent results when dealing with multi-objective problems. Based on Azizi M, et al [27], Multi-Objective Atomic Orbital Search (MOAOS) is proposed to solve multi-objective optimization problems. By using MOAOS to evaluate benchmark problems ZDT and DTLZ, the results show that this algorithm can produce superior or similar results compared with other meta-heuristic methods. Khodadadi N, Abualigah L, Mirjalili S [28] made appropriate changes to the stochastic paint optimizer (SPO) and proposed a multi-objective stochastic paint optimizer (MOSPO) to solve multi-objective optimization problems. Compared with MOPSO MSSA and MOALO, MOSPO had high convergence and excellent Pareto front results in dealing with multi-objective engineering problems.

Reviewer #2:

Comment to the Authors

 The readability and presentation of the study should be further improved. The paper suffers from language problems. It is suggested to proofread by a native speaker or a proofreading agent.

Response

Thank you for your careful review. The manuscript has been thoroughly revised and rewritten by a native English speaker.

 The importance of this study is not clear. The authors are suggested to state the reason for proposing this algorithm.

Response

Thank you for your suggestions. As suggested by reviewer, we add the importance of this paper and the reason for the proposed algorithm in the manuscript (Page 6, lines 131-134). The details are as follows:

It is not difficult to conclude from the above literature review that TCQT has been widely addressed and effectively resolved. However, the existing TCQT mathematical model has not considered the impact of reward and punishment factors on construction projects, and intelligent optimization algorithms with better solution performance need to be proposed.

 It is recommended to revise the contribution of this study by highlighting its novelty and finding of this study.

Response

Thank you for your suggestions. As suggested by reviewer, we have revised the contributions of this study in the manuscript (Page 6, lines 134-145). Our reply is as follows:

Thereby, the contributions of this study are stated as follows: (1) clarify the nonlinear relationship between time and bonus-penalty cost, a new bonus-penalty cost model is presented. Therefore，a new multi-objective mathematical model of the time-cost quality trade-off problem with several equality and inequality constraints is established. (2) A new multi-objective immune wolf swarm algorithm has been developed to solve the trade-off problem of time, cost, and quality. The proposed MOIWCA can solve the problem that the traditional wolf colony algorithm is easy to fall into a local optimal solution. At the same time, in order to improve the search speed and optimization performance of the algorithm, the cross operation and immune operation in the immune algorithm are improved. The algorithm can provide a representative and managed Pareto set for the time-cost-quality trade off problem. (3) To make sure that the proposed MOIWCA works and can be used, we use a high-speed railway construction project as a case study to show that the proposed method is better.

 This study suffers from a lack of deep literature review on new and improved metaheuristic algorithms such as Migration-based moth-flame optimization algorithm, Hybridizing of whale and moth-flame optimization algorithms to solve diverse scales of optimal power flow problem, An improved moth-flame optimization algorithm with adaptation mechanism to solve numerical and mechanical engineering problems, and GGWO: Gaze cues learning-based grey wolf optimizer and its applications for solving engineering problems.

Response

We are extremely grateful to you for pointing out this problem. The references you provided are very consistent with the research content, and have high academic value, which is very helpful to our paper. Therefore, we decided to use the two articles you suggested to further enrich our references in the manuscript (Page 5, lines 109-113 and 117-122). The details are as follows：

Nadimi-Shahraki, et al [24] proposed an improved moth-flame optimization (I-MFO) algorithm, introduced the adapted wandering around search (AWAS) strategy to escape the local optimal solution, evaluated the performance of the proposed algorithm through a benchmark function, and compared it with other well-known metaheuristic algorithms. Finally, I-MFO is used to solve practical mechanical engineering problems. To Nadimi-Shahraki MH, et al [26] reduce the high selection pressure and low diversification of GWO algorithm, a grey wolf optimizer based on gaze cue learning (GGWO) is proposed. The algorithm introduced two strategies: neighbor gaze cues learning (NGCL) and random gaze cues learning (RGCL)to enhance the optimization ability. The GGWO algorithm is compared with other algorithms. The results show that the GGWO algorithm has better performance. 

 It is recommended to consider the flowchart or pseudocode of the proposed algorithm.

Response

Thank you for your suggestions. In the paper, we have provided the flow chart of the proposed algorithm, that is, Fig 8. (Flowchart of MOIWCA for the fuzzy TCQT problem) (Page 25, line 500).

Fig 8 Flowchart of MOIWCA for the fuzzy TCQT problem

 It is recommended to experimentally report the weak points of the immune wolf colony algorithm. then clarify how the proposed algorithm can overcome these weak points to solve the time-cost-quality trade-off problem.

Response

Thank you for your suggestions. As suggested by reviewer, in order to better illustrate the advantages of the algorithm in dealing with the time cost quality tradeoff problem, we modified and supplemented the algorithm design part of the paper (Pages 19-21, lines 392-416). The details are as follows:

The immune wolf colony hybrid algorithm proposed in this paper focuses on the combination of wolf colony algorithm and immune algorithm, which makes the hybrid algorithm have the advantages of immune algorithm's excellent solving performance in combinatorial optimization and wolf colony algorithm's fast convergence in solving problems, and avoids the disadvantages of immune algorithm's slow convergence speed in optimization problems and wolf colony algorithm's easy falling into local extremum. Therefore, the immune wolf colony hybrid algorithm provides a new way to solve the time-cost quality trade-off (TCQT) problem.

When using the immune wolf colony hybrid algorithm to solve the TCQT problem, this paper takes the individual wolf colony as the antibody of the immune algorithm, the odor concentration of prey as the antigen of the immune algorithm, and the odor concentration of the individual wolf colony as the fitness value of the current solution. The process of wolf colony searching, and trapping prey is to use immune wolf colony hybrid algorithm to solve the TCQT problem iteratively. In the process of wolf pack updating, it is always hoped that the wolf with high adaptability will be left behind. However, if the superior wolf is too concentrated, it is difficult to ensure the diversity of the whole wolf colony. Therefore, by using the mechanism of antibody concentration inhibition, the antibody with low affinity and high concentration will be suppressed, and the antibody with high affinity and low concentration will be retained and promote the production, to ensure the diversity of antibody groups. 

The basic principle of IWCA is to combine the immune principle of antibody concentration inhibition mechanism and immune memory function in the immune algorithm with the wolf colony algorithm, calculate the concentration of wolves(antibody) and compare it with the initial wolf concentration (Cu). If it is higher than Cu, update the wolves using IWCA, otherwise, update the wolves using the self-adapting WCA. Self-adapting WCA can accelerate search efficiency. The immune operation increases wolf colony diversity, ensures convergence speed, and improves the global search ability and accuracy. 

 Equations 36 and 37 are not clear. Please clarify them.

Response

Thank you for your suggestions. We are sorry that we clearly did not express Equations 36 and 37. Equation 36 is based on equation 29, and uses the adaptive parameter adjustment in particle swarm optimization to improve rush factor θ (Page23, lines 459-471). The details are as follows:

a. Perform dynamic self-adapting of calling behavior of the wolf colony. For solving the local optimal solution of WCA, by referring to the adaptive parameter adjustment method in the particle swarm optimization algorithm, change the rush factor θ in Eq. (29) into a dynamic weight coefficient θ^*, to change the fixed step of the calling behavior. In this paper, dynamic adaptive calling behavior of WCA algorithm is proposed. The new calling behavior of the wolf colony according to the following equations:

 x_id^(k+1)=x_id^k+θ^* (x_ad^k-x_id^k ) 〖step〗_b (36)

 θ^*={█(θ_min^*+(θ_max-θ_min )(F_avg^k-F_min^k )/(F_id^k-F_min^k ) F_id^k≥F_avg^k@θ_max^* F_id^k<F_avg^k )┤ (37)

where θ^* is dynamic adaptive coefficient, θ_max^* and θ_min^* are the maximum and minimum inertial coefficients respectively; F_min^k is the minimum value of fitness function of wolf colony in the k generation; F_id^k is the value of fitness function of the wolf i in the d dimensional space when the k generation; F_avg^k is the average value of fitness function

 It is recommended to boost the visualization of this study by increasing the experimental evaluations.

Response

Thank you for your suggestions. We know that the supplement of experimental evaluations will improve the overall level of our paper, but the main purpose of the paper is to apply the proposed IWCA to solve the engineering problems of the Beijing Shanghai high-speed railway construction project, and the results show that the method is effective. However, we add the proposed IWCA comparison experiment between IGPSO, NSGA-II and MOEA/D. The comparison results are shown in Table 1 and Table 2. (Pages 26-27, lines 513-541) We will incorporate your suggestions into our future work, meanwhile, based on your suggestions, add performance comparison experiments with other intelligent algorithms to further evaluate the proposed algorithms in the future.

Table 1. Comparison of Test Results of Rosenbrock Function Using IWCA, IGPSO, NSGA-II and MOEA/D

 Average running time (ms) Average iteration generations Mean error Standard deviation Success rate (%) Best minimum value

IGPSO 8,687 2,183 0.429618 0.256765 100.00 0.007728

IWCA 7,568 1,750 0.415738 0.235836 100.00 0.004258

NSGA-II 15,366 3,628 0.820524 0.446531 100.00 0.008795

MOEA/D 11,082 2,825 0.646358 0.315873 100.00 0.007936

Table 2. Comparison of Test Results of Rastrigin Function Using IWCA and IGPSO

 Average running time (ms) Average iteration generations Mean error Standard deviation Success rate (%) Best minimum value

IGPSO 18,046 3,928 0.367794 0.241865 100.00 0.004157

IWCA 9,658 2,486 0.285682 0.224783 100.00 0.003265

NSGA-II 27,863 6,838 0.824563 0.468735 100 0.013648

MOEA/D 22,648 5,082 0.693425 0.336749 100 0.008543

 It is recommended to determine the claims of this study and which experimental evaluation can support them.

Response

Thank you for your suggestions. We know that the supplement of experimental evaluations will improve the overall level of our paper, but the main purpose of the paper is to apply the proposed WOIWCA to solve the engineering problems of the Beijing Shanghai high-speed railway construction project, and the results show that the method is effective. As the response to question 8, through the comparative experiment with IGPSO, NSGA-II and MOEA/D, the proposed IWCA performs excellent in average running time, average iteration generations, mean error, standard deviation, success rate and best minimum value. We will incorporate your suggestions into our future work, meanwhile, based on your suggestions, add performance comparison experiments with other intelligent algorithms to further evaluate the proposed algorithms and determine the claims in the future.

 The discussion section should be added in a more highlighting, argumentative way. The author should analyze the reason why the tested results are achieved.

Response

Thank you for your suggestions. As suggested by reviewer, not only we improved the analysis of Fig. 9, but also enriched the discussion and analysis of experimental results by adding Figs. 10-12. (Pages 30-31, lines 593-613). The specific analysis contents are as follows:

The non-dominated solutions may also be used to optimize tradeoffs between any two objectives on a two-dimensional plane. Figs. 10-12 show the trade-off between time and cost, time and quality, and cost and quality, respectively. As illustrated in Fig 10, if the investment in the construction project is increased, the duration of completing the project will be shortened, and vice versa. As shown in the Fig 11, if the duration of the construction project is increased, the quality of the completed project will increase first and then decrease. The situation shows that excessive extension of the construction project will lead to a decline in the quality of the project. One of the possible reasons for the situation is that long hours of work will reduce the working efficiency of labors. As shown in the Fig 12, if the quality of construction projects is improved, the investment in construction projects will increase. However, blindly increasing investment will not make the quality higher and higher but can reduce the quality. One of the possible reasons for the situation is that the purpose of increasing costs is to shorten the duration of the construction project, and the reduction of time leads to the decline of the quality of the construction project.

Fig.10. Time-cost tradeoff analysis

Fig.11. Time-quality tradeoff analysis

Fig.12. Cost-quality tradeoff analysis

 The single-point crossover operation based on the crossover of points with the same serial number is not clear. Please increase the understanding and impact of this operator in the proposed algorithm.

Response

Thank you for your suggestions. As suggested by reviewer, we re-describe the proposed single point crossover operation (Page 23, lines 475-482). The details are as follows: 

In this paper, a single point crossover operation based on the same gene number. The difference between the proposed method and the classical single point crossover operation is that the selection of crossover points is not random. The specific method is to randomly select two chromosomes, select the same gene number from the two parent chromosomes as the intersection point, and then split and exchange the remaining chromosomes on the right side from this point to obtain new offspring. The new crossover operation method can improve the running speed of the algorithm and ensure the diversity of the population. A single point crossover operation based on the same gene number is shown in Fig 6.

Fig 6. single point crossover operation based on the same gene number

 The statistical evaluation analysis is recommended.

Response

Thank you for your suggestion. As suggested by reviewer, we select SP and HV for analysis (Page 31-33, lines 615-652). The details are as follows：

We compared MOIWCA performance against NSGA-II, MOPSO and MODE to assess comparative effectiveness. For comparison purposes, all four algorithms used an equal number of function evaluations, had a population size of 300, and a maximum of 500 generations. In NSGA-II, the crossover probability is set at 0.5 and the mutation probability is set at 0.1. In MOPSO, the two learning factors c1, c2 are both chosen at 1.5, and the inertia factor w is set in the range of 0.2–0.6. The MOIWCA control parameters remained the same, as stated previously in Table 3. Thirty independent runs are carried out for all experiments. 

Multi-objective optimization problem performance measures are more complex than those of single-objective optimization problems. Three issues are typically considered: (1) convergence to the Pareto optimal set; (2) preservation of diversity in Pareto optimal set solutions; and (3) the maximal distribution bound of the Pareto optimal set [38]. In the literature, the researchers have suggested numerous quality indicators. In this paper, the Spacing metric, and the Hyper volume (HV) metric are selected to evaluate the performance of different algorithms.

(1) Spacing metric: This indicator assesses the degree of distribution of non-dominated solutions. The mathematical definition of the spacing metric may be given as:

∆=(∑_(i=1)^m▒〖d(E_i,Ω)+∑_(X∈Ω)▒|d(X,Ω)-d ® | 〗)/(∑_(i=1)^m▒d(E_i,Ω) +(|Ω|-m)d ® ) (40)

where Ω is a set of non-dominated Pareto-front solutions, (E_1,E_2,…,E_k) are k extreme solutions in the set is true Pareto-front, m is the number of objectives and d(X,Ω)=min¦(Y∈Ω,Y≠X) ‖F(X)-F(Y)‖ is the minimum Euclidean distance between solution X and its neighboring solutions in the obtained non-dominated X set, d ®=1/|Ω| ∑_(X∈Ω)▒d(X,Ω) is the mean value of all d(X,Ω), |Ω| is the total solutions in Ω set. A value of 0 indicates that all members of Pareto optimal solution are equally spaced. The smaller the value, the better distribution and diversity of non-dominated solutions. Table 6 shows a comparison of the Spacing metric for different algorithms. 

Table 6. Comparison of Spacing metric for different algorithm

 MOIWCA MODE MOPSO NSGA-II

Best 0.4336 0.4453 0.4627 0.4852

Worst 0.6132 0.7025 0.9416 0.9653

Average 0.5235 0.5431 0.7123 0.7356

Standard Deviation 0.0397 0.0586 0.1542 0.1653

(2) Hyper-volume (HV): This indicator calculates the volume (in the objective space) covered by members of a non-dominated set of solutions Ω to a problem that works to minimize all objectives. A hypercube v_iis constructed for each solution X_i∈Ω, with reference point W and the solution Xi as the diagonal corners of the hypercube. The reference point may be found simply by constructing a vector of the worst objective function values. Thereafter, a union of all hypercubes is found, with the HV of this union calculated as:

HV=⋃_(i=1)^|Ω|▒v_i (41)

Algorithms with larger HV values are desirable. The HV value of a set of solutions is normalized using a reference set of Pareto optimal solutions with the same reference point. After normalization, the HV values are confined to range [0,1]. Table 7 lists the results for each of the four compared algorithms in terms of HV. From Table 7, we can see that the proposed model gets the highest HV values. This means that the MOIWCA has better convergence and diversity performance than the other algorithms.

Table 7. Comparison of HV metric for different algorithm

 MOIWCA MODE MOPSO NSGA-II

Best 0.9486 0.8965 0.9284 0.8652

Worst 0.7164 0.6932 0.4683 0.2473

Average 0.8207 0.7648 0.7136 0.5241

Standard Deviation 0.0519 0.0546 0.1483 0.2062

 It is recommended to increase the understanding of Figs 5 and 9.

Response

Thank you for your suggestion. As suggested by reviewer, For Fig 5, we have added our understanding of the figure (Page 16, lines 318-320). The details are as follows:

Fig. 5 shows that SDi, BDi, LDi are the shortest duration, best duration, longest duration of activity i, and BDi corresponds to the maximum value of QPI. The parameters can be provided by the deterministic data from the case study.

Fig.5. The curve of the relationship of time- quality

For Fig. 9, we have improved the analysis of the Fig 9 (Pages 29-30, lines 581-592). The details are as follows:

Fig 9 shows the typical Pareto optimal fronts obtained using the MOIWCA for this case study. These three fronts clearly show the relationships among project duration, cost, and quality. It is not difficult to see that the relationship between time, cost and quality is non-linear. For example, the project manager can appropriately extend the time and reduce the cost according to the actual situation of the construction project, and the quality can meet the requirements of the construction project; Similarly, the quality can also meet the requirements of the construction project by appropriately reducing the time but increasing the cost. However, the quality of the construction project cannot meet the minimum requirements if the cost is constantly increased by blindly seeking to reduce the time of the construction project or blindly increasing the time to reduce the cost. This three-dimensional representation of the tradeoffs may help decision-makers figure out how the different possible plans for using resources will affect the performance of the project.

Fig. 9. Time–cost–quality tradeoff Pareto front using MOIWCA.

 The figures before equations 7-10 are not clear.

Response

Thank you for your suggestion. As suggested by reviewer, we adjusted the size of the figures in equations 7-10, we hope it can meet the reviewer’s standard (Page 10-11, lines 216, 217, 222 and 223).

 The nomenclature table is recommended for the parameters used in this study.

Response

Thank you for your suggestion. As suggested by reviewer, we modify the Table 3, and the new Table 3 is shown as follow (Page 28, line 559).

Table 3. Best MOIWCA parameter for 14 activities in the Beijing-Shanghai high speed railway project.

Parameters Significance Value

ND Population size 112

T Number of generations 300

Pc Crossover probability 0.85

Pm Mutation probability 0.055

μ Expected reproduction index 0.95

stepa Wandering step 0.9

stepb Calling step 0.6

stepc Siege step 0.5

λ Constant of expected reproduction 0.3

We tried our best to improve the manuscript and made some changes in the manuscript. These changes will not influence the content and framework of the paper. And here we did not list the changes but marked in red in revised paper. We appreciate for Editors/Reviewers’ warm work earnestly, and hope that the correction will meet with approval. Once again, thank you very much for your comments and suggestions.

---

## [Decision Letter · Decision Letter 1]

2 Nov 2022

PONE-D-22-22367R1Multiple objective immune wolf colony algorithm for solving time-cost-quality trade-off problemPLOS ONE

Dear Dr. Li,

Thank you for submitting your manuscript to PLOS ONE. After careful consideration, we feel that it has merit but does not fully meet PLOS ONE’s publication criteria as it currently stands. Therefore, we invite you to submit a revised version of the manuscript that addresses the points raised during the review process.

We look forward to receiving your revised manuscript.

Kind regards,

Seyedali Mirjalili

Academic Editor

PLOS ONE

Journal Requirements:

Reviewers' comments:

Reviewer's Responses to Questions

**Comments to the Author**

1. If the authors have adequately addressed your comments raised in a previous round of review and you feel that this manuscript is now acceptable for publication, you may indicate that here to bypass the “Comments to the Author” section, enter your conflict of interest statement in the “Confidential to Editor” section, and submit your "Accept" recommendation.

Reviewer #1: All comments have been addressed

Reviewer #2: (No Response)

2. Is the manuscript technically sound, and do the data support the conclusions?

Reviewer #1: Yes

Reviewer #2: Partly

3. Has the statistical analysis been performed appropriately and rigorously? 

Reviewer #1: Yes

Reviewer #2: Yes

4. Have the authors made all data underlying the findings in their manuscript fully available?

Reviewer #1: Yes

Reviewer #2: Yes

5. Is the manuscript presented in an intelligible fashion and written in standard English?

Reviewer #1: Yes

Reviewer #2: Yes

6. Review Comments to the Author

Reviewer #1: Thank you for your consideration.

Reviewer #2: Thank the authors for responding to comments however, some comments have still remained and should be responded to. Thus, befor accepting this manuscript for publishing, responding to the following minor comments is needed.

1. It is recommended to experimentally report the weak points of the immune wolf colony algorithm. then clarify how the proposed algorithm can overcome these weak points to solve the time-cost-quality trade-off problem.

2. Thank the authors for reviewing more literature but the limitations of already-existing metaheuristic algorithms are not addressed, nor is explained the need to develop an enhanced variant of an optimization algorithm. Thus, more new and improved metaheuristic algorithms such as the following studies should be considered. Starling murmuration optimizer: A novel bio-inspired algorithm for global and engineering optimization, QANA: Quantum-based avian navigation optimizer algorithm, DMDE: Diversity-maintained multi-trial vector differential evolution algorithm for non-decomposition large-scale global optimization.

3. It is recommended to make sense of results gained by different algorithms and their differences for the spacing metrics using Eq. (40).

4. It is recommended to boost the visualization of this study by increasing the experimental evaluations.

5. The conclusion section also needs significant revisions. It should be linked to research questions, and address the research contributions.

7. PLOS authors have the option to publish the peer review history of their article (what does this mean?). If published, this will include your full peer review and any attached files.

Reviewer #1: **Yes: **Nima Khodadadi

Reviewer #2: No

---

## [Author Response · Author response to Decision Letter 1]

13 Nov 2022

1. It is recommended to experimentally report the weak points of the immune wolf colony algorithm. then clarify how the proposed algorithm can overcome these weak points to solve the time-cost-quality trade-off problem.

Response

Thank you for your suggestion. As suggested by reviewer, we add the weak of the IWCA, and explain how the MOIWCA perform the TCQT problem in the manuscript (Pages 26-27, lines 507-529). The details are as follows:

In the above content, based on grey wolf optimizer (GWO) algorithm [41] and immune mechanism, we proposed immune wolf colony algorithm (IWCA). It can find the best and unique solution when dealing with single objective problems, but IWCA cannot solve multi-objective optimization problems. The reason is due to the sub objectives of the multi-objective optimization problem are contradictory, and the improvement of one sub objective may cause the performance of another or several other sub objectives to change accordingly (such as the TCQP problem to be solved in this paper). That is, it is impossible to achieve the optimal value of multiple sub objectives at the same time, but only to coordinate and compromise among them, make each sub goal as optimal as possible. Its essential difference from the single objective optimization problem is that the solutions of the multi-objective optimization problem are not unique, but there is a set of optimal solutions composed of many Pareto optimal solutions. Each element in the set is called Pareto optimal solution or non-inferior optimal solution. In order to preform multi-objective optimization problems by IWCA, we combine the idea of multi-objective optimization with the excellent strategy of IWCA, and propose a multi-objective immune wolf colony algorithm (MOIWCA). We learn from the components used in MOPSO [42] and MOGWO [43], and integrate two new components on the basis of IWCA. The first component is to create an archive, which is responsible for storing all non-dominated Pareto optimal solutions generated during algorithm iteration. When the optimal solution is to be archived or the archive is full, it can be controlled through the archive controller. The second component is the leader selection strategy, which is used to select from files α wolf, β wolves, and γ Wolves is the leader in the hunting process. This component can select the least crowded part of the search space in the best archive, and provide one of the non-dominant solutions. The selection is done by roulette-roulette method. These two components have been described in detail in the literature [43], and we will not repeat in the paper.

2. Thank the authors for reviewing more literature but the limitations of already-existing metaheuristic algorithms are not addressed, nor is explained the need to develop an enhanced variant of an optimization algorithm. Thus, more new and improved metaheuristic algorithms such as the following studies should be considered. Starling murmuration optimizer: A novel bio-inspired algorithm for global and engineering optimization, QANA: Quantum-based avian navigation optimizer algorithm, DMDE: Diversity-maintained multi-trial vector differential evolution algorithm for non-decomposition large-scale global optimization.

Response

Thank you for your suggestions. we have added these papers in the manuscript (Page 5, lines 102). Our reply is as follows:

At present, common optimization algorithms include particle swarm optimization algorithm, fast non-dominated sorting genetic algorithm with elite retention strategy (NSGA-II), ant colony algorithm, simulated annealing algorithm, and so forth [22-24].

The references (22-24) are advised by reviewer.

3. It is recommended to make sense of results gained by different algorithms and their differences for the spacing metrics using Eq. (40).

Response

Thank you for your suggestions. As suggested by reviewer, we have revised in the manuscript (Page 34, lines 663-667). Our reply is as follows:

The smaller the value of the Spacing metric, the better distribution and diversity of non-dominated solutions. From the comparison results of the Spacing metric for different algorithms, the MOIWCA obtained the minimum Spacing metric. This shows that in dealing with multi-objective optimization problems, The non-dominated solutions obtained by MOIWCA have better distribution and diversity than other algorithms.

4. It is recommended to boost the visualization of this study by increasing the experimental evaluations.

Response

Thank you for your suggestions. In the paper, we add Fig 13 and Fig 14 to improve the visualization of this paper.

Fig 13. Comparison of Spacing metric for different algorithm.

Fig 14. Comparison of HV metric for different algorithm.

5. The conclusion section also needs significant revisions. It should be linked to research questions, and address the research contributions.

Response

Thank you for your suggestions. As suggested by reviewer, we revise the conclusion in the manuscript (Page 39, lines 689-705). Our reply is as follows:

In this paper, the time-cost-quality trade-off problem is studied, and a hybrid algorithm MOIWCA is developed to solve the TCQT problem for construction project. The paper makes three important contributions: first, based on the traditional time-cost model, we introduce the bonus-penalty mechanism, and propose a new nonlinear time-cost model. Meanwhile, we develop a new QPI model by analyzing the qualitative and quantitative relationship between time and quality. Second, we propose a multi-objective immune wolf colony algorithm (MOIWCA)to solve the TCOQ problem. The design of algorithm is divided into two parts: we propose IWCA that combines the wolf colony algorithm and immune algorithm, and improves the crossover operation and mutation operation. In addition, we integrate IWCA with multi-objective optimization to enable it to deal with time-cost-quality trade-off problem. Thirdly, we take the Beijing-Shanghai high-speed railway construction project as a case to evaluate the performance of the MOIWCA. The results show that the MOIWCA is more effective and efficient than widely used multi-objective algorithms (MOPSO, NSGA-II, MODE). The SP and HV values comparisons indicated that the proposed MOIWCA performs excellent in terms of distribution, convergence and uniformity.

Results show that the proposed method MOIWCA generates a better Pareto front than widely used approaches. With the obtained non-dominated solutions, the project managers can easily trade-off the three important objectives, including project time, cost, and quality.

---

## [Editor Report · Decision Letter 2]

21 Nov 2022

Multiple objective immune wolf colony algorithm for solving time-cost-quality trade-off problem

PONE-D-22-22367R2

Dear Dr. Li,

We’re pleased to inform you that your manuscript has been judged scientifically suitable for publication and will be formally accepted for publication once it meets all outstanding technical requirements.

Kind regards,

Seyedali Mirjalili

Academic Editor

PLOS ONE
---

## [Editor Report · Acceptance letter]

16 Dec 2022

PONE-D-22-22367R2 

Multiple objective immune wolf colony algorithm for solving time-cost-quality trade-off problem 

Dear Dr. Li:

I'm pleased to inform you that your manuscript has been deemed suitable for publication in PLOS ONE. Congratulations! Your manuscript is now with our production department. 

Kind regards, 

on behalf of

Prof. Seyedali Mirjalili 

Academic Editor

PLOS ONE